# MODELING FUTURE CONVERSATION TURNS TO TEACH LLMS TO ASK CLARIFYING QUESTIONS

**Michael J.Q. Zhang[♥,♦], W. Bradley Knox[♦], Eunsol Choi[♥,♦]**

New York University[♥], The University of Texas at Austin[♦]

michaelzhang@nyu.edu

## ABSTRACT

Large language models (LLMs) must often respond to highly ambiguous user requests. In such cases, the LLM's best response may be to ask a clarifying question to elicit more information. Existing LLMs often respond by presupposing a single interpretation of such ambiguous requests, frustrating users who intended a different interpretation. We speculate this is caused by current preference data labeling practice, where LLM responses are evaluated only on their prior contexts. To address this, we assign preference labels by simulating their expected outcomes in future turns. This allows LLMs to learn to ask clarifying questions when it can generate responses that are tailored to each user interpretation in future turns. On open-domain QA datasets with multiple annotations, we evaluate systems based on their ability to ask clarifying questions to recover each user's interpretation and expected answer. We compare systems trained using our proposed preference labeling methods against standard methods, which assign preferences based on only prior context. Our method achieves a 5% improvement in F1 measured against the answer set from different interpretations of each query, showing the value of modeling future conversation turns. We further demonstrate that our method can be used to train models to judiciously determine when to ask clarifying questions, directly answering the question when clarification is unnecessary. In our experiments, we find that our method achives a 3% improvement in accuracy of such judgments over existing methods.

## 1 INTRODUCTION

Ambiguity is a hallmark of natural language that enables concise communication by allowing speakers to exclude details that are inferable from the context (e.g., conversational, temporal, geographical) (Piantadosi et al., 2012). At times, however, the speaker's intent is unclear despite the context, and further interaction is necessary to clarify their intent. Asking clarifying questions is particularly important for large language models (LLMs), which are tasked with serving a wide audience, often without access to the personalized context available in human interactions. In this work, we develop LLMs that can ask clarifying questions to resolve ambiguity in their users' requests.

State-of-the-art LLMs (OpenAI, 2023; Gemini Team, 2024) often do not ask clarifying questions when presented with an ambiguous request, and instead respond directly by assuming the user's intent (see an example in Figure 1). We speculate that this tendency stems from a shortcoming in their RLHF training pipelines, which utilize annotated preference data (Ouyang et al., 2022). In standard preference data collection, annotators are given a conversation history and are tasked with ranking options for the next assistant turn (Bai et al., 2022; Wang et al., 2024). These annotation schemes only consider preferences over single-turns of interaction, making it difficult for annotators to assess the utility of a clarifying question. Furthermore, this can lead to biases where annotators prefer responses with complete but presumptuous answers over incomplete clarifying questions.

We propose an alternative method for annotating *double-turn* preferences over clarifying questions where annotators interact with the LLM by providing their responses to clarifying questions and observing the LLM's subsequent responses. Each annotator then assigns preferences based on whether the completed interaction successfully fulfilled the request. When comparing multiple clarifying

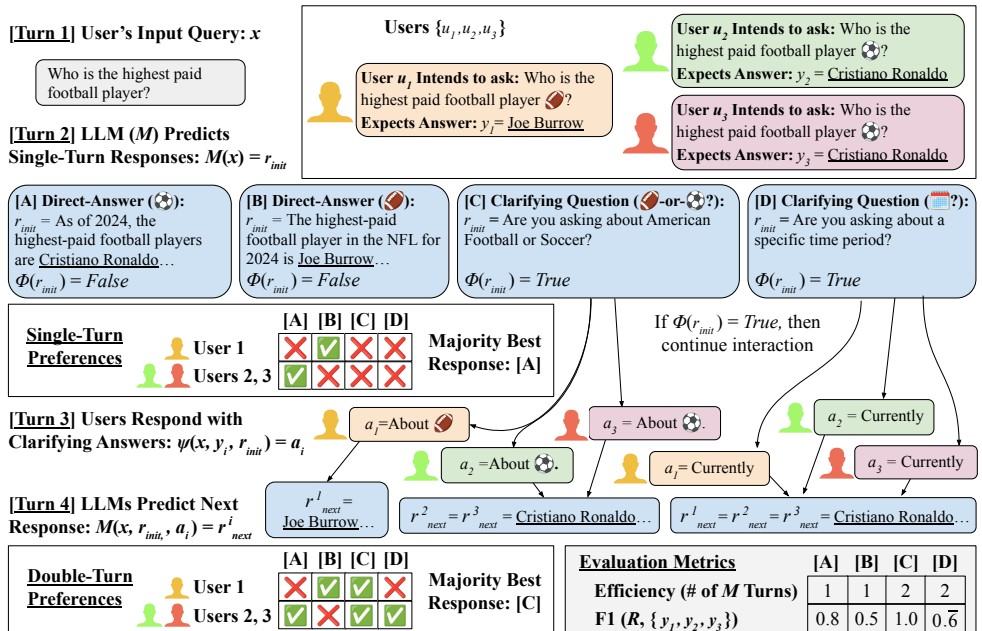

Figure 1: Our interaction scenario and preference labeling schemes. We aim to build an LLM that can interact with user to generate the final answer set $R$, containing an answer for each user, for the input query $x$. In this example, we include two responses from state-of-the-art LLMs ([A] from GPT-4 and [B] from Gemini, full responses in Appendix A), which both presuppose an interpretation of the word football. We also include two clarifying responses ([C] and [D]) where [C] correctly disambiguates the two intended interpretations across all users. We depict two ways to assign preferences on LLM's initial output, single-turn and our proposed double-turn.

questions and direct answer responses, each annotator's preferences are aggregated to identify the response that maximizes preference across all annotators. We depict our proposed double-turn preference annotation method and compare against standard methods in Figure 1.

To demonstrate the benefits of training LLMs to ask clarifying questions with double-turn preferences, we experiment on open-domain QA (Chen et al., 2017). We develop an automatic evaluation framework for evaluating clarifying questions that use simulated user interactions (Section 2). We evaluate systems on two axes: *efficiency* measured by the number of model turns and *effectiveness* measured by F1 score between the predicted answer set for each user and their expected gold answer.

Our experimental results show that training systems with double-turn preferences outperforms training with standard preferences annotation methods (Zhu et al., 2023), resulting in consistent 4-5% improvement in F1 score over three base LLMs (Dubey et al., 2024; Mesnard et al., 2024). We also demonstrate that double-turn preferences can be used to train systems that determine if clarification is needed, or if the user's question can be answered without additional interaction.

We summarize our contributions below:

- We establish an automatic framework for evaluating double-turn interaction involving clarifying question. Our framework includes a user simulation model and automatic evaluation metrics measuring the system's ability to predict outputs that are tailored to each user efficiently.
- We develop a *double-turn* preference annotation method for labeling preferences over clarifying question and direct response that utilizes the outcomes in later turns of interaction.
- Our results on Open-Domain QA datasets (Kwiatkowski et al., 2019; Min et al., 2020) demonstrate that training systems to ask clarifying questions using our double-turn preferences improves QA performance over training with standard single-turn preferences. We also show that training with *double-turn* preferences can teach models to judiciously decide when to ask for clarification.

Our evaluation paradigm and preference labeling scheme can further support future research in developing interactive LLM assistants. We release all code and data at https://github.com/mikejqzhang/clarifying_questions.

## 2 TASK: QA WITH SIMULATED USER INTERACTION

We chose to study open-domain QA, as it is both prevalent in real user-assistant interactions (Zhao et al., 2024) and the scope of necessary clarification questions is not immediately identifiable from the input or the task. For simplicity, we choose a well-studied short factoid QA setting (Kwiatkowski et al., 2019; Min et al., 2020) where an answer to the question consists of a few tokens. Below, we describe our task and evaluation metrics. Figure 1 contains an example of task scenario.

### 2.1 INTERACTION SCENARIO

**Setting / Goal:** We assume an input query $x$, which can be potentially ambiguous, and a set of $k$ users $u_1, ...u_k$ who posed the input query $x$. Each $i$-th user $u_i$ has an disambiguated intent for query $(x_i)$ and an expected gold answer $(y_i)$. Our goal is to build a system $M$ that returns an answer set $R$ that can satisfy as many users as possible (by providing the expected answer $y_i$ for each user $u_i$) with the minimum number of interactions (measured by the number of model turns).

We will have a simple function $\phi(M(x))$ that returns *True* if $M(x)$ is a clarifying question, and *False* otherwise. We further assume an oracle user simulation model $\psi(x, r_{init}, y_i) = a_i$, that takes first turn input $x$, model's initial output $M(x)$, and $i$-th user's targeted answer $y_i$, that generates a response to the model's initial response $M(x)$.

**Dialogue Turns:** In the initial turn, user poses the input query $x$. In the second turn, given the input query $x$, the system $M$ is to generate its initial response ($r_{\text{init}}$). Depending on whether $r_{\text{init}}$ is a question, it will trigger different dialogue trajectories.

- **Double-turn Scenario** If the initial response is a clarifying question ($\phi(r_{\text{init}})$=*True*), we continue the conversation with the user simulation model. For each of $i$-th user, the evaluation system generates a third turn ($a_i$) with the user simulation model $\psi(x, r_{\text{init}}, y_i) = a_i$. For the fourth turn, the system will take each of the $k$ interaction traces $(x, r_{\text{init}}, a_i)$ to generate the fourth turn response for $i$-th user $M(x, r_{\text{init}}, a_i) = r_{\text{next}}^i$. The final output answer set from the system will be a set of $k$ answers $R = \{r_{\text{next}}^1, r_{\text{next}}^2, ...r_{\text{next}}^k\}$, where each $r_{\text{next}}^i$ is a response for $i$-th user.
- **Single-turn Scenario** Otherwise if $\phi(r_{\text{init}})$=*False*, we will consider the model output $R = \{r_{\text{init}}\}$ as the final prediction. To allow a model that does not generate clarifying questions to generate multiple answers, we sample multiple answers from the model, up to $k$ answers. The final answer set in this case will be $R = \{r_{\text{init}}^1, ...r_{\text{init}}^k\}$.

Existing QA models mostly generates an single answer to the initial input query $x$, even when the query is ambiguous. Such systems that predict an answer without first confirming the user's intent may mislead users with incorrect answers and fail to serve the entirety of their user population. In this work, we explore methods for *QA-with-Clarification*, where models can ask the user a clarifying question $q$ and observe the user's response $a$ before predicting an answer. A system can also sometimes poses a clarifying question and sometimes directly answer (Zhang & Choi, 2023a).

**Evaluations:** We evaluate the system in two axis: efficiency and effectiveness. To measure efficiency, we simply measure the average number of total turns from the system. For unambiguous inputs, asking clarifying question is unnecessary. If the system can satisfy users without incurring additional interaction, such system should be preferred.

To measure effectiveness, we use **F1** between the model generated answer set ($R$) and the gold reference answer set ($\{y_1, ..y_k\}$). When evaluating QA-with-Clarification systems, we additionally enforce that answers must be correct for their corresponding clarifying answer. We hold QA-with-Clarifications to this higher standard, as it aligns with our goal of to developing systems that can identify different user interpretations of a query and accurately predict corresponding answers.

### 2.2 EVALUATION FRAMEWORK IMPLEMENTATION

**Data:** We will leverage existing open-domain QA datasets (Kwiatkowski et al., 2019; Min et al., 2020) where each query is paired with annotated answers from multiple annotators.

**Identifying clarifying question** $\phi(\cdot)$**:** We will use a very simple method, which takes the input string and check whether it starts with "Clarifying Question:".

**User Simulation Model $\psi$ for Turn 3:** We use GPT-4 as our oracle user simulation model ($\psi$). We prompt GPT-4 with the input query $x$, clarifying question $q$, and target answer $y_i$ and task it with generating a corresponding clarifying answer ($(x, q, y_i) \rightarrow a_i$). We prompt GPT-4 to abstain from providing a clarifying answer $a_i$ if it judges that none exists. This happens when the clarifying question does not address the ambiguity in the query, hence we count the resulting target answer prediction $r^i_{\text{next}}$ as incorrect. We further ensure that the generated clarifying answers do not leak the target answer by removing instances where the target answer $y_i$ appears in the predicted clarifying answer $a_i$, treating such examples the same as abstains from GPT-4 (full prompt in Appendix C).

## 3  FINE-TUNING LLMS TO ASK CLARIFYING QUESTIONS

We present our approach to build a model that can engage with users for a multi-turn interaction, ask clarifying question when it can satisfy more users in the future turns. Following the standard RLHF training pipeline (Ouyang et al., 2022), we first construct an instruction tuning dataset focusing on clarifying responses as desired outputs. We then use this constructed dataset for supervised finetuning (SFT), before performing preference learning between two possible responses. We first describe our method for constructing this instruction tuning dataset before introducing our annotation scheme for assigning preferences over clarifying questions.

### 3.1  PREFERENCE DATA GENERATION FROM USER-LLM INTERACTIONS

Our method for labeling preferences deviates from standard annotation methods (Bai et al., 2022; Wang et al., 2024) in two ways. First, we simulate an additional interaction with the user and derive reward from the LLM's final response after the additional turn. Second, we expect multiple annotators for each example. By using multiple human annotators to identify a set of expected answers, we simulate each annotator's interactions with the LLM and their preferences. While prior work has explored using multiple annotator judgments in single-turn preference datasets (Wang et al., 2024; Kopf et al., 2023), these works aggregate annotator judgments via majority choice to generate a single preference label per example. This practice, however, can lead to adverse outcomes where systems whose outputs are catered toward a single, majority accepted response (Fleisig et al., 2023; Santy et al., 2023). In contrast, we use preferences across multiple annotators without removing individual annotator judgments, helping us identify whether a clarifying question or direct-answer response successfully accommodates the range of different annotator's interpretations.

In our annotation scheme (depicted in Figure 2), annotators are provided an input query $x$ with several candidate clarifying question $q$ and direct-answer responses $\widehat{y}$. They respond to each clarifying question by providing the clarifying answer that corresponds to their interpretation. We then show annotators the model's final output prediction, given each clarification. Annotators then assign their preferences over clarifying questions and direct-answer responses based on whether the final output matched their expected interpretation. We then aggregate these preferences to identify which response was preferred by the most annotators.

To prevent systems from asking unnecessary clarifying questions, ties between direct-answer and clarifying question responses are broken by favoring the direct-answer response. The final aggregated preferences can then be used in conjunction with standard RL methods from human preferences for training. While this annotation scheme can be applied to human annotators, in this work we use simulated user interactions for annotation, simulating different users' interactions from existing datasets with multiple annotations (discussed below in Section 4).

**Simulating User Interaction**  To simulate different user interpretations of the input query, we utilize datasets consisting of an input query $x$ and a set of correct answer outputs $\{y_1, ..., y_k\}$. For each query, we use the set of gold outputs to simulate $k$ different users, one for each answer in $\{a_1, ..., a_k\}$ which are identified by multiple human annotators (we discuss how these annotations are sourced in Section 4). We then simulate the behaviors of each of these users and their responses to a proposed clarifying question $q$ using a trained *user-simulator* model, which takes as input query $x$, clarifying question $q$, and the user's annotated answer $y_i$ and is trained to predict the user's clarifying answer $(x, q, y_i) \rightarrow a_i$. While we use prompted GPT-4 to simulate clarifying answers from users for evaluation, during training we use this trained user-simulator model to reduce costs and to ensure that our systems do not overfit to a particular user-simulator method.

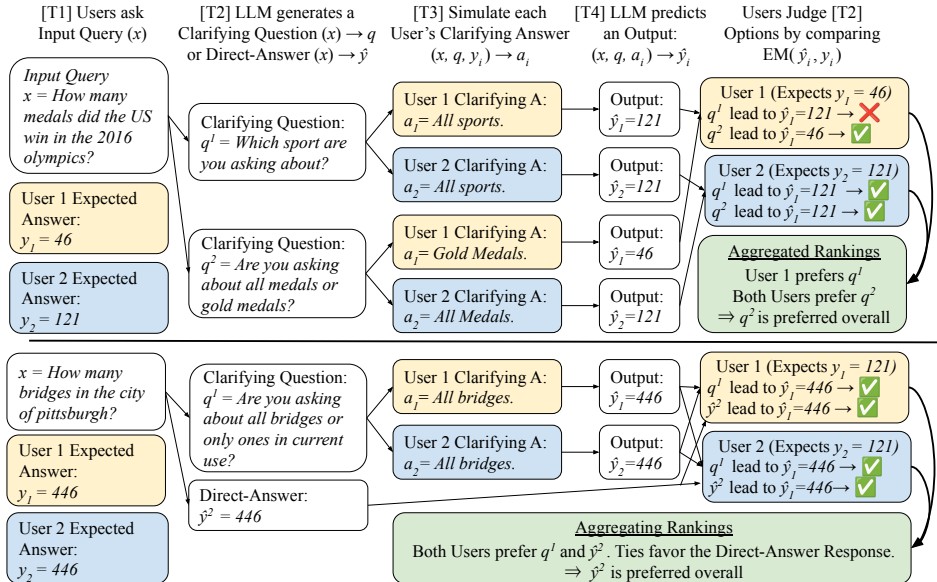

Figure 2: Depiction of our preference annotation method. Here, simulated users provide their responses to model-generated clarifying questions and determine preference based on which clarifying question or direct-answer responses lead to their expected answer. We then aggregate preferences across users by selecting the response that is preferred by the most users while minimizing the number of user interactions turns.

**Answer Prediction / Assigning Preferences** Conditioning on the simulated clarifying answer, our LLM model predicts the output $\widehat{y_i}$, $(x, q, a_i) \rightarrow \widehat{y_i}$. As a last step, we aggregate annotator preferences so that clarifying questions are ranked by their EM accuracy, evaluated against each annotator's expected answer: $\sum_{i=1}^{k} \text{EM}(\widehat{y_i}, y_i)/k$. To rank direct-answer responses, which only predict a single answer $\widehat{y}$, we similarly evaluate the EM accuracy of the predicted answer $\widehat{y}$ against each annotator's expected answer. To prevent systems from asking unnecessary clarifying questions, we favor the direct-answer response in the case of ties. We then use these aggregated rankings to determine the pairwise preference labels between responses for training.

## 4 EXPERIMENTAL SETTINGS

**Models** We use Llama2-7b (Touvron et al., 2023), Gemma-7b (Mesnard et al., 2024), and Llama3-8b (Dubey et al., 2024) as our base LLMs. We forego using the instruction-tuned models due to data leakage concerns, as such systems are finetuned on the entirety of NQ-Open. For training and inference, we use 8-bit quantization (Dettmers et al., 2022) with LoRA (Hu et al., 2022; Dettmers et al., 2024) (training details in Appendix C).

**Data** We perform our experiments on the NaturalQuestions (NQ-Open) (Kwiatkowski et al., 2019; Lee et al., 2019) and AmbigQA (Min et al., 2020) datasets. In both of these datasets, each input query $x$ are associated with a set of possible output answers $\{y_1, \ldots, y_k\}$. We describe each dataset below, and provide details in Appendix A.

- **NQ-Open** is comprised of questions from Google search queries by real users. Each query is annotated with answers from Wikipedia by up to five annotators; many questions (about 10% of train and 42% of development set) in the dataset contain multiple distinct answers based on each annotator's judgment in interpreting the query and selecting the best answer span. The answers sets identified by annotators for such examples can stem from a variety of ambiguities, from formatting to ambiguities resulting in semantically distinct answers (examples in Appendix A).
- **AmbigQA** additionally annotates a subset of NQ-Open whether the question is ambiguous. If the input is ambiguous, annotators then provide the additional answers to other possible interpretations of the query. This process identifies about 56% of all queries in NQ-Open as ambiguous, and recovers 2.6x more answers to ambiguous questions.

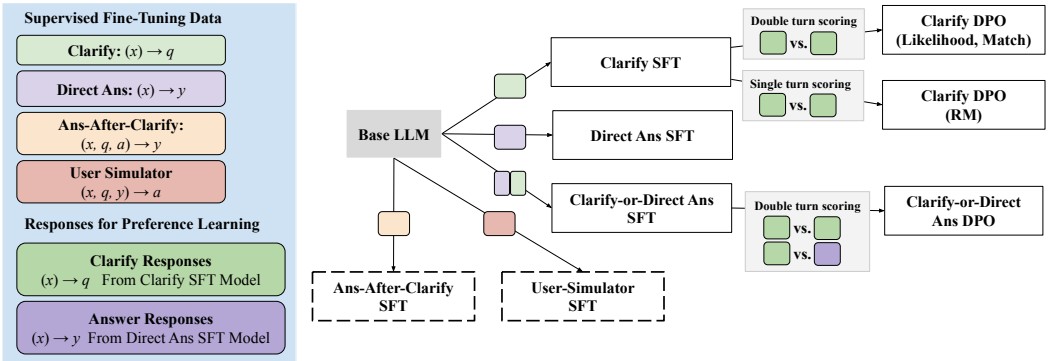

Figure 3: Illustration of models in our study (right) and the data used for training them (left). Ans-After-Clarify SFT model is used to generate responses for the fourth turn, User-simulator SFT model is used to generate responses for the third turn. All other models generate responses for the 2nd turn.

We evaluate on AmbigQA test set (n=1960). In addition to evaluating over the **Full** test set, we additionally report performance on splits of **Unambiguous** questions (788 questions with one answer each) and **Ambiguous** questions (1,172 questions averaging 3.7 answers each).

**Supervised Finetuning (SFT) Dataset Generation** To generate examples for SFT training, we prompt GPT-4 with an input query $x$ and a set of feasible answers $Y = \{y_1, \ldots, y_k\}$ and task it with proposing a clarifying question $q$ that will help determine which answer the questioner expects. Our prompt also tasks GPT-4 with predicting responses to the clarifying question for each one of the feasible answers $\{(a_1, y_1), \ldots, (a_k, y_k)\}$, if it determines one exists. We use the same GPT-4 prompt to construct all examples SFT training, and introduce two methods for generating feasible answer sets $Y$ for a given query $x$.

- **Answers from Human Annotations:** We use the ambiguous queries from the AmbigQA training and development splits and their annotated answers as the feasible answer set $Y_{\text{human}}$. Identifying ambiguities and labeling all possible answers is a challenging task for annotators, and thus this data can be expensive to collect. Furthermore, we hypothesize that what is ambiguous for an LLM often deviates from what is ambiguous for humans.
- **Answers from Model Predictions:** We build feasible answer sets $Y_{\text{model}}$ for queries in the NQ-Open training set using answer candidates proposed by a base LLM. Prior work has demonstrated that base LLMs are well calibrated in open-domain QA tasks with few-shot prompting (Kadavath et al., 2022; Cole et al., 2023). Drawing on these observations, we construct a feasible answer set $Y_{\text{model}}$ by sampling 5-shot predictions from Llama-2-7b (Touvron et al., 2023). For each query, we sample 5-shot prompt and generate its greedy answer with temperature $T = 0.0$ and sample an answer with $T = 1.0$. We repeat the process 10 times per query and remove repeated answers to generate a set of model-identified feasible answers. We further filter examples where none of the feasible answers match any annotated answers from NQ-Open.

For each method of generating feasible answer sets ($Y_{\text{human}}$ and $Y_{\text{model}}$), we generate an SFT dataset of 4,400 input query and clarifying question pairs $(x, q)$, which we split into training (4,000) and development (400) splits. Between both datasets, this gives us a total of 19,807 $(x, q, a_i, y_i)$ examples. In Appendix A, we include examples and the exact prompts.

## 4.1 COMPARED SYSTEMS

Figure 3 summarizes all models used in our experiments. We have two types of models, model that will be used at the second turn (which we refer as Turn 2 model) and model that will be used at fourth turn (which we refer as Turn 4 model and notate as "Ans-After-Clarify" model in the figure). The model that will be used at the second turn can ask clarification question or directly answer the question. All models that are capable of asking clarification questions are paired with the same Turn 4 model, which is trained over all generated SFT examples. We use a separate model for performing this latter step to ensure that differences in performance only reflect the value of the

clarifying question, and explore training joint models for both steps in Section 5. We describe each of these below.

**Direct Ans SFT** This model is trained to predict the answer given the input query $(x) \rightarrow y$. We fine-tune base LLM with next-token prediction loss on the full set of question answer pairs (i.e., each $(x, y_i)$) in our generated SFT training sets. We use this model under two inference settings: **(1) Greedy** where we take the single greedy answer prediction and **(2) Sampled** where we sample a set predicted answers from the LLM. We sample 20 answers from the model and select the top $k$ most frequent answers, where $k$ matches the gold number of annotated answers.[1]

**Clarify SFT** This model is trained to ask clarifying question to input query. We finetune base LLMs on our SFT datasets constructed with both human annotations and model predicted answers. In Section 5, we provide ablation experiments training over each dataset to compare their utility.

**Clarify DPO (RM, Likelihood, Match)** These models are each fine-tuned from Clarify SFT model with using DPO (Rafailov et al., 2023), a method for optimizing LLMs on preference data. It uses the following loss for updating the policy LLM $\pi_\theta$ based on the base reference policy $\pi_{ref}$, prompt $h$, and preferred/rejected responses $(r_p, r_r)$:

$$L(\pi_\theta, \pi_{ref}, r_p, r_r, h) = \log \sigma(\beta \log \frac{\pi_\theta(r_p|h)}{\pi_{ref}(r_p|h)} - \beta \log \frac{\pi_\theta(r_r|h)}{\pi_{ref}(r_r|h)})$$

where $\beta$ is a hyperparameter and $\pi(r|h)$ is the predicted probability of generating $r$ given the prompt $h$. These models are provided the same response pairs sampled from Clarify SFT model,[2] only differing in how reward are assigned.

- **RM**: This uses off-the-shelf reward model to rank clarifying questions. We use the Starling-RM-7B-alpha reward model (Zhu et al., 2023) which was finetuned on over 3.8M single-turn preferences between the outputs from a variety of LLMs.
- **Likelihood**: In this method, clarifying questions are ranked by the summed likelihoods of generating each gold answer after observing the simulated users' clarifying answers. Using likelihood to score questions has been proposed in prior work studying clarifying question generation to identify different user personas (Andukuri et al., 2024).
- **Match**: We rank clarifying questions by whether the predicted answer **matches** each user's gold answer after observing each simulated user's clarifying answer, averaged over all simulated users.

**Clarify-or-Direct Ans SFT / DPO** We train a Clarify-or-Direct Ans SFT model on the union of all data used to train our Clarify SFT and Direct-Ans SFT models above. To train our Clarify-or-Direct Ans DPO model, wefurther train our Clarify-or-Direct Ans SFT model above using double-turn preferences over clarifying question and direct-answer responses with the DPO learning objective. To rank responses, we use answer matching as depicted in Figure 2. We perform DPO training over pairwise comparisons from the same six candidate clarifying questions generated for our Clarify DPO models above and the single greedy sampled direct-answer response generated from our Direct Ans SFT model above.

## 5 RESULTS

We report our main results in Table 1, capturing the number of user turns in the conversation (# ↓) and Answer F1. Overall, systems that are equipped to ask clarifying questions outperform Direct-Ans models in effectiveness (Answer F1). Clarify-or-Answer methods strike a balance between effectiveness and efficiency.

Training with our double-turn preference labeling scheme consistently achives the best answer F1, with our match method yielding the best results overall. We observe mixed results from training with single-turn RLHF annotations from the Starling RM: yielding positive results with Llama2 as the base LLM and minor increases/decreases in performance for Gemma and Llama3 when evaluated over both ambiguous and unambiguous queries. These results support our claim that single-turn

---

[1] In rare cases (< 2%), there are less than $k$ unique answers within the 20 samples.

[2] We sample six candidate clarifying questions (one with $T = 0.0$ and five with $T = 1.0$) and form preference pairs among them.

Table 1: Main results, separated by systems that never clarify (# Turns = 1), always clarify (# Turns = 2), or can do both (# Turns $\in [1, 2]$). The shaded blocks represent systems that simulate future turns. Our method (Clarify DPO w/ Match) yields the optimal results in Answer F1 at the cost of always asking clarifying questions. Our method that selectively asks clarifying questions (Clarify-or-Direct-Ans DPO strikes a balance between efficiency (# Turns) and effectiveness (Answer F1). Bold results denote statistically significant differences from all other methods with the same base model (over $N = 10,000$ samples with $p < 0.01$).

| | # ($\downarrow$) | Llama2 Answer F1 ($\uparrow$) Unamb / Amb / All | # ($\downarrow$) | Llama3 Answer F1 ($\uparrow$) Unamb / Amb / All | # ($\downarrow$) | Gemma Answer F1 ($\uparrow$) Unamb / Amb / All |
|---|---|---|---|---|---|---|
| Direct-Ans SFT | | | | | | |
| w/ Greedy | 1 | 25.4 / 16.8 / 21.1 | 1 | 31.2 / 19.2 / 24.8 | 1 | 26.1 / 16.8 / 21.1 |
| w/ Sampled | 1 | 25.0 / 17.2 / 21.4 | 1 | 28.2 / 20.2 / 24.7 | 1 | 23.7 / 17.9 / 21.4 |
| Clarify SFT | 2 | 31.0 / 21.6 / 25.9 | 2 | 37.6 / 26.5 / 31.5 | 2 | 35.7 / 23.6 / 28.8 |
| Clarify DPO | | | | | | |
| w/ RM | 2 | 31.0 / 25.7 / 28.3 | 2 | 36.2 / 26.7 / 30.9 | 2 | 33.9 / 25.7 / 29.5 |
| w/ Likelihood | 2 | 30.2 / 23.9 / 27.2 | 2 | 43.5 / 29.6 / 359 | 2 | 37.3 / 26.8 / 31.5 |
| w/ Match | 2 | **38.3 / 28.2 / 32.8** | 2 | 42.9 / **3.17 / 36.5** | 2 | **40.7 / 28.6 / 33.9** |
| Clarify-or-Direct-Ans | | | | | | |
| SFT | 1.12 | 25.6 / 18.4 / 21.3 | 1.40 | 35.3 / 23.5 / 28.2 | 1.43 | 22.3 / 19.0 / 20.3 |
| DPO | 1.56 | 28.9 / 21.1 / 24.3 | 1.57 | 35.2 / 25.1 / 29.1 | 1.61 | 28.2 / 22.2 / 24.6 |

Table 2: Answer F1 results comparing training Clarify DPO (Ours) systems using SFT training on clarifying questions generated from $Y_{\text{human}}$ or $Y_{\text{model}}$ answer sets or both.

| SFT Data | Llama2 (7b) | | | Llama3 (7b) | | | Gemma (7b) | | |
|---|---|---|---|---|---|---|---|---|---|
| | Unamb | Amb | All | Unamb | Amb | All | Unamb | Amb | All |
| Human | 28.8 | 23.6 | 25.9 | 40.1 | 30.9 | 34.9 | 37.9 | 27.4 | 32.3 |
| Model | 29.9 | 26.4 | 28.3 | 42.3 | 31.2 | 36.4 | 39.1 | 28.3 | 32.9 |
| Both | 38.3 | 28.2 | 32.8 | 42.9 | 31.7 | 36.5 | 40.7 | 28.6 | 33.9 |

RLHF annotation struggles to identify useful clarifying questions. In contrast, training with double-turn preferences, where clarifying questions are assessed based on their later outcomes, demonstrates consistent improvements in the LLM's ability to generate useful clarifying questions, on both ambiguous and unambiguous queries. This demonstrates that clarifying questions not only help models disambiguate user intents for ambiguous queries, but also can help models recover correct answers in general even for unambiguous queries. We include such example generations in Appendix D.

**Ablations: Comparing Clarifying Question Generated with Human Annotated Answers vs. Model Predicted Answers** So far, we performed SFT training on the union of clarifying questions generated from model-identified ($Y_{\text{model}}$) and human-identified ($Y_{\text{human}}$) feasible answer sets. Is one of them more useful than the other? In Table 2, we compare performance using SFT examples constructed from our $Y_{\text{model}}$ and $Y_{\text{human}}$ answer sets, after RLHF training with double-turn preference.

With Llama2 base model, we see clear improvements when using our model-ambiguity SFT dataset over using our human-ambiguity dataset alone. These improvements even hold when evaluated against the answer sets from AmbigQA, despite the fact that our human-ambiguity SFT dataset was generated to distinguish between answers labeled in the AmbigQA training split. However, on Gemma and LLama3, we find only minor improvements from training with model-ambiguity SFT dataset. The model-ambiguity dataset was using feasible answer sets $Y_{\text{model}}$ from Llama2 predictions. This gap suggests that constructing a model-ambiguity dataset that is specific to the base model improves the resultant clarifying question quality.

**A Joint Model for Asking Clarifying Questions and Predicting Answers** Here, we experiment with using a single LLM for both the clarification ($x \rightarrow q$) and answer ($x, q, a_i \rightarrow y_i$) conversation turns. To create this joint model, we take inspiration from prior works demonstrating that averaging the weights of multiple finetuned models is a simple and effective approach to joining the capabilities of finetuned mod-

Table 3: Answer F1 results comparing separate versus joint clarifying question and answering models (Cl / Ans).

| | Llama2 (7b) | | | Llama3 (7b) | | | Gemma (7b) | | |
|---|---|---|---|---|---|---|---|---|---|
| | Unamb | Amb | All | Unamb | Amb | All | Unamb | Amb | All |
| Separate | 38.3 | 28.2 | 32.8 | 42.9 | 31.7 | 36.5 | 40.7 | 28.6 | 33.9 |
| Joint | 32.4 | 25.0 | 28.7 | 41.0 | 29.2 | 34.6 | 36.2 | 27.4 | 31.4 |

els into a single LLM (Wortsman et al., 2022). We merge the LoRA parameter updates from our clarification model (trained with preferences from full interactions) and the answer prediction models used in our main results (Table 1).

Table 3 compares the joint model against using separate LLMs for each conversation turn. Overall performance degrades when using a joint model for both turns; however, we are still able to retain most of the performance gains over Direct-Ans and QA-with-Clarification methods trained with single-turn preferences. Future work may explore alternative training methods that mitigate the performance degradation, such as using mixture-of-experts methods for LLMs (Jiang et al., 2024) or improved multi-turn learning algorithms (Zhou et al., 2024b).

## 5.1 DETERMINING WHEN CLARIFYING QUESTIONS ARE NECESSARY

We further investigate our system's ability to identify when clarifying questions are necessary. To evaluate our Clarify-or-Answer model's predictions, we report **Direct-Answer Accuracy**: Did the the system correctly decide to directly answer the question when the question was unambiguous and the greedy predicted direct-answer was correct. To determine whether the greedy predicted direct-answer was correct, we decode our Clarify-or-Answer system's greedy direct-answer prediction for each input question. We also report **Ambig Acc**: the accuracy of a system's clarify-vs-answer predictions on the human ambiguous-vs-unambiguous labels.

We compare our models predictions (C-or-A-Pred) against a **random** baseline and two additional methods, **ProCot** and **PPDPP**. To compute the **random** baseline, we fix the percent of direct-answer responses (DA%) to our model's clarify-or-answer predictions and randomly sample DA% of predictions to directly answer, and predict clarifying questions for the remaining queries. **ProCot** Deng et al. (2023a) utilizes a CoT prompt to determine whether or not the input is ambiguous and a clarifying

Table 4: Additional results for Clarify-or-Answer method (Llama2 model). We include the percent of questions directly answered (DA%), and evaluate using F1, Direct-Answer (DA) Acc., and Ambig (A) Acc.

| Method (DA%) | Ans F1 ↑ | DA Acc ↑ | A Acc ↑ |
|---|---|---|---|
| Random (44%) | 23.4 | 55.4 | 52.1 |
| Ours (44%) | **24.3** | **61.9** | 53.7 |
| Ours w/ ProCoT (35%) | 15.6 | 60.1 | 49.2 |
| Ours w/ PPDPP (46%) | 23.9 | 57.7 | **59.0** |

question is necessary. In practice, we use the instruction-tuned variants of each base model to make these predictions, as we find irregular behavior when applying the CoT prompt to non-instruction tuned models. **PPDPP** Deng et al. (2023b) proposes a method for training an auxiliary model to predict how models should respond. Here, we train an additional copy of the base model make the binary prediction of whether asking a clarifying question is necessary for the system recover more all answers. **ProCot** and **PPDPP** do not train models to what questions to ask, and are only designed to predict whether clarifying is necessary. To evaluate these baselines, we pass their predictions to our trained Clarify-or-Answer model, which then generate the clarifying question of direct answer according to the baseline's predicitons. We include implementation details in Appendix B.

We report our results with Llama2 in Table 4 (results with other base models in Appendix B). Our Clarify-or-Answer system's predictions consistently outperforms all baselines across all metrics except Ambig Accuracy. Despite using a stronger base model (instruction tuned variant instead of base model), the prompt-based method (ProCot Prompt) performs the worst, as it is not trained in-domain. Training a separate model for ambiguity prediction (PPDPP) performs well, showing almost 10% higher ambiguity accuracy than our full model. However, our method still outperforms this model in answer accuracy. This demonstrates that our method (1) more frequently asks clarifying questions when it's useful to do so; and (2) avoids asking for clarification on ambiguous questions when the model is not able to successfully use the clarifying question. The relatively large gains on Direct-Answer accuracy compared to F1 indicate that we can expect performance on end-task metrics to further improve with better clarifying question generation.

## 6 RELATED WORK

**Ambiguity in NLP** Ambiguity has been studied extensively in NLP in across a variety of tasks. In machine translation, works have studied instances where sentences in a source language have multiple valid translations in a target language (Fernandes et al., 2021; Lopes et al., 2020; Voita et al.,

2019). In natural language inference (NLI), Nie et al. (2020) collected a dataset of highly cross-annotated examples that contain high disagreement in annotator judgments. Liu et al. (2023) then found that these disagreements are often the result of ambiguity in the input. Recent work has studied various sources of ambiguity, from entity-linking (Lee et al., 2024), co-reference ambiguities (Yuan et al., 2023), to temporal and geographical contexts (Zhang & Choi, 2021).

**Uncertainty in LLMs** Several recent works have studied calibration in LLMs, looking specifically at the effects of RLHF training. Specifically, whereas likelihoods for base LLMs (i.e., pretrained) tend to be well-calibrated for a variety of tasks (Cole et al., 2023; Kadavath et al., 2022), this behavior is lost after RLHF finetuning (OpenAI, 2023). Additionally, Zhou et al. (2024a) investigated uncertainty-expression via epistemic markers generated by LLMs and found evidence that standard single-turn RLHF training encourages over-confidence in LLM responses. Band et al. (2024) trained an LLM to generate long-form text that includes confidence statements about its own generation. Zhang & Choi (2023b) studied when uncertainty in predictions are caused by lack of knowledge or by lack of clarification.

**Clarifying Question** Prior works for training LLMs to ask clarifying questions have focused on settings where the types of clarification required are fixed by the task (i.e., there is fixed set of relevant input features (Wei et al., 2018; Kuhn et al., 2022)) or the input (e.g., reading-comprehension tasks where the context contains multiple correct answers (Guo et al., 2021)). Such works have studied clarifying question generation for a variety of tasks, including gauging social and moral situations (Pyatkin et al., 2022) and image classification (Yu et al., 2019).

Closely related to our work, Andukuri et al. (2024) trained LLMs to ask clarifying questions to resolve task ambiguity, where a single request may have multiple different indented tasks depending on the user, after user simulation. Compared to their experimental setting of simulated user persona for open-ended tasks, we address real-world open domain question answering task. Furthermore, we leverage preference optimization framework while they focus on generating supervised fine-tuning data by filtering low-quality clarifying questions. Hong et al. (2023) also uses LLM-simulated dialogues for training. Recent work has also noted the scarcity of clarifying questions in responses from existing LLMs, and have studied promoting methods for eliciting such responses with greater frequency (Deng et al., 2023a; Shaikh et al., 2023). Li et al. (2023) found that clarifying questions can be useful for learning individualized preferences.

## 7 LIMITATIONS AND FUTURE WORK

We only consider systems for one or two turn interactions and does not consider cases where asking additional clarifying questions might be helpful after observing the user's clarifying answer. Future work might explore methods for extending our evaluation framework and double-turn preference annotation methods to accommodate general multi-turn interactions with users. Furthermore, the systems explored in this work also do not model dialogue acts (Stolcke et al., 2000) outside of predicting a single answer or asking a clarifying question. At times, it might be more appropriate for LLMs to generate responses for different behaviors (e.g., Overton responses containing multiple answers if there are only a few possible interpretations. Long vs. short answers for complex queries. Abstaining in lieu of predicting erroneous answers). Future work might consider how clarifying questions should be used in conjunction with other strategies. Another important direction is extending double-turn preference method for other tasks. Our evaluation framework is intended to be easily adaptable, only requiring (1) multiple annotator labels for expected outputs and (2) an changing end-task evaluation metric for the new setting.

## 8 CONCLUSION

We propose a method for training LLMs to ask clarifying questions with double-turn preferences and an an automatic framework for evaluating systems that ask clarifying questions using user simulated user interactions. Our QA results demonstrate that training systems to ask clarifying questions with double-turn preferences improves performance over training with standard single-turn preference labels and can be used to train models to judiciously decide when to ask for clarification.

ACKNOWLEDGMENTS

We thank the UT NLP community for providing feedback throughout the project. This work was in part supported by Cisco Research. Any opinions, findings and conclusions, or recommendations expressed in this material are those of the authors and do not necessarily reflect the views of Cisco Research. The work is partially supported by a gift from Google Research. This work has taken place in part in the Rewarding Lab at UT Austin. The Rewarding Lab is supported by NSF (IIS-2402650), ONR (N00014-22-1-2204), EA Ventures, Bosch, UT Austin's Good Systems grand challenge, and Open Philanthropy. This work was done in part while the first and last author was visiting the Simons Institute for the Theory of Computing.

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

## A    DATASET DETAILS

In Table 5, we provide examples from AmbigQA and NQ-Open with differing answer sets. These examples highlight instances where relying on a multiple annotators to provide the answer versus relying on a single annotator to provide all possible answers yielded differing answer sets.

### A.1    HUMAN AND MODEL AMBIGUITY SFT AND RLHF TRAINING DATASET DETAILS

In Table 8, we report dataset statistics for or two SFT datasets. Note that $(x, q)$ pairs are used for SFT training for our clarifying question asking $(x \rightarrow q)$ systems. Additionally, the $(x, q, a, y)$ pairs are used for training our joint and separate systems for predicting QA answers given clarification $(x, q, a \rightarrow y)$ as well as our user simulator models $(x, q, y \rightarrow a)$. We also use these pairs for training our direct QA systems $(x \rightarrow y)$, ignoring the clarifying question and answer. In Table 9, we include several examples from each dataset, highlighting the differences in the ambiguities identified in each.

For RLHF training, we use examples from the NQ-Open training set after removing examples used to generate our SFT datasets from NQ-Open. This leaves us with 70,904 remaining input questions, which we split into training and develpent splits (64,584 and 6,320).

## B    ADDITIONAL EXPERIMENTS

### B.1    CLARIFY-OR-ANSWER BASELINES

Here, we describe the additional baselines introduced in Section 5.1.

**Proactive chain-of-thought (ProCoT) prompt** Deng et al. (2023a) presents a chain-of-thought prompt method for instructing LLMs to first determine whether or not an input question is ambiguous before generating a direct-answer or clarifying question response. The need for instruction-following abilities for this baseline necessitates using the instruction-tuned variants of each base model. Therefore, performance of this approach may be inflated due to the stronger base model.

**Plug-and-Play Dialogue Policy Planner (PPDPP)** Deng et al. (2023b) proposes a SFT and RL methods for training a separate model predicting the proper dialogue act for the next assistant turn. Because their method is not designed to train systems to generate responses, only to decide which type or response to predict, we train a separate PPDPP model from our base LLM via SFT. This system is trained over the SFT training sets over the gold dialogue act: clarify if the question is ambiguous or directly answer otherwise. We do not include their RL method, as the authors report comparable performance with and without additional RL training. We found training to be hyper parameter sensitive, with the suggested hyperparameter training systems to always select a single option. Using this trained system, we then use its predictions to determine whether to predict a clarifying question or direct-answer response using our Clarify-or-Direct Ans DPO model.

### B.2    ADDITIONAL RESULTS

In Table 6, we report the additional results comparing different methods for determining when clarifying questions are necessary with additional base models (Llama3 and Gemma). We find that the ProCot Chain-of-Thought prompt yields degenerate behavior when used with the instruction tuned Gemma and Llama3 models, failing to ever predict to directly answer a question. When looking at the other methods, we find similar trends to the results from Llama2 discussed in Section 5.1.

## C    EXPERIMENTAL DETAILS

We perform all experiments on a single machine with 8 A40 (48GB) GPUs using the transformers library (Wolf et al., 2020) and the AdamW optimizer (all training runs completed within 24 hours). During SFT training, we train all models with a learning rate of 5e-5 and batch size of 32. Training was performed for up to 5 epochs, evaluating on our development set after each epoch and selecting using the best performing checkpoint.

| Question | NQ-Open Annotations Answers | AmbigQA Annotations Is Ambiguous? | Answers |
|---|---|---|---|
| Who is singing in the background of gold digger? | {"Ray Charles", "Jamie Foxx"} | No | {"Jamie Foxx"} |
| How much does a 2004 chevy cavalier weight? | {"1,162 - 1,315 kg", "2,562 - 2,900 lb" } | No | {"2,562–2,900 lb"} |
| What season does meredith and derek get married in grey's anatomy? | {"Season 7"} | Yes | {"Season 5", "Season 7"} |
| When did the soviet union test its first atomic bomb? | {"29 August 1949"} | Yes | {"29 August 1949", "7:00 a.m"} |
| When did the soviet union test its first atomic bomb? | {"29 August 1949"} | Yes | {"29 August 1949", "7:00 a.m"} |

Table 5: QA examples differing labeled answer sets form NQ-Open and AmbigQA.

Table 6: Llama3 Results for Clarify-or-Answer methods. For each model, we include the percent of questions directly answered (DA%), and evaluate using F1, Direct-Answer Acc., and Ambig Acc.

| Base Model | Method (DA%) | Ans F1 | Direct-Answer Acc | Ambig Acc |
|---|---|---|---|---|
| Llama3 | Random (43%) | 24.5 | 55.6 | 51.1 |
| | Ours (43%) | **25.1** | **59.2** | **54.0** |
| | PPDPP (60%) | 22.2 | 45.3 | 53.4 |
| Gemma | Random (38%) | 23.6 | 60.0 | 52.1 |
| | Ours (38%) | **24.6** | **64.1** | 54.3 |
| | PPDPP (54%) | 21.7 | 50.8 | **56.7** |

For DPO training, we merge LoRA checkpoints from our SFT-only baseline and train using a KL regularization factor of 0.1 and a learning rate of 5e-6 in all experiments. For training was performed for up to 2 epochs until loss converges on development data, selecting the best performing checkpoint on development data. For Llama2-7b based methods, we train with a batch size of 32, evaluating every 750 steps. For Gemma-7b based methods we train with a batch size of 16, evaluating every 1500 steps.

For performing additional SFT training to create our joint models for asking clarifying questions and answering with clarification, we merge LoRA checkpoints again and use the same hyperparameters used for standard SFT training listed above.

## D EXAMPLE OUTPUTS

We include examples of models successfully using clarifying questions to predict the target answers for ambiguous (in Table 14) and unambiguous test queries (in Table 13).

## E LICENSING

The transformers library is licensed under Apache License 2.0. AmbigQA does not list any license; however NaturalQuestions, the dataset which it is based on, is under Apache License 2.0. Llama2 is licensed under the LLAMA 2 Community License Agreement. Gemma is licensed under the Gemma Terms of Use. We license all our created assets under CC BY-NC-SA 4.0.

| **Input Query:** Who is the highest paid football player? |
|---|

**GPT-4 Response (104 words):**
As of 2024, the highest-paid football players are Cristiano Ronaldo and Karim Benzema, both earning around €200 million per year. Ronaldo plays for Al Nassr, and Benzema is with Al Ittihad, both in the Saudi Pro League.

Following them, N'Golo Kanté and Neymar each earn approximately €100 million annually, also playing in the Saudi Pro League for Al Ittihad and Al Hilal, respectively. Kylian Mbappé is the highest-paid player outside the Saudi Pro League, with a salary of €70 million at Paris Saint-Germain.

These figures reflect the significant financial influence of the Saudi Pro League, which has attracted many top players with lucrative contracts.

**Gemini Response (30 words):**
The highest-paid football player in the NFL for 2024 is Joe Burrow, the quarterback for the Cincinnati Bengals. He earns a whopping $55 million per year according to his contract.

**Claude-3 Sonnet Response (112 words):**
As of 2023, the highest paid football (soccer) player in the world is Kylian Mbappé of Paris Saint-Germain. His reported annual salary is around €72 million (approximately $78 million). Some other top highest-paid football players in 2023 include:

2. Lionel Messi (Paris Saint-Germain) - Around €41 million per year
3. Neymar (Paris Saint-Germain) - Around €36.5 million per year
4. Kevin De Bruyne (Manchester City) - Around €25 million per year
5. Robert Lewandowski (Barcelona) - Around €23.5 million per year

However, player salaries can fluctuate frequently with new contracts, bonuses and endorsements. Mbappé's new deal with PSG from 2022 made him the highest paid at the reported €72 million annual wages.

Table 7: LLM responses to the input query. All responses were generated on May 21st, 2024.

| SFT Dataset | Train | | Dev | |
|---|---|---|---|---|
| | $(x, q)$ | $(x, q, a, y)$ | $(x, q)$ | $(x, q, a, y)$ |
| Human-Identified Ambiguity | 4000 | 8781 | 400 | 889 |
| Model-Identified Ambiguity | 4000 | 9189 | 400 | 948 |

Table 8: SFT Dataset Statistics.

| Question ($x$) | Clarifying Question ($q$) | Clarifying Answers ($a_i$) | Answers ($y_i$) |
|---|---|---|---|
| where were the olympic games held in greece | Are you asking about the ancient Olympic Games or the modern Olympic Games? | Ancient Olympic Games / Modern Olympic Games | Olympia / Athens |
| who won the red army or the white army | Are you asking about the Russian Civil War or a specific battle? | Russian Civil War / Specific battle | The Red Army / the white army |
| who wrote ob la di ob la da | Are you asking about the primary composer or the co-writer? | Primary composer / Co-writer | Paul McCartney / John Lennon |
| original singer of how am i supposed to live without you | Are you asking about the original singer who first recorded the song or the singer who popularized it? | The original singer who first recorded the song / The singer who popularized the song | Laura Branigan / Michael Bolton |

Table 9: Randomly sampled Model-Ambiguity SFT dataset examples.

| Question ($x$) | Clarifying Question ($q$) | Clarifying Answers ($a_i$) | Answers ($y_i$) |
|---|---|---|---|
| Who played thanos in gardians of the galaxy? | Are you asking about the actor who provided the voice for Thanos, or the actor who physically portrayed him on screen? | The actor who provided the voice for Thanos. / The actor who physically portrayed him on screen. | Isaac C. Singleton Jr. / Sean Gunn |
| Where in the bible is the story of samuel? | Are you asking for the specific book and verses where the story of Samuel is found, or are you asking for the general section of the Bible where the story is located? | I'm asking for the specific book and verses. / I'm asking for the general section of the Bible. | 1 Samuel 1:1–28 / Hebrew Bible |
| Most amount of points scored in an nba game? | Are you asking about the most points scored by a single player in a game or the highest combined score in a game? | Most points scored by a single player / Highest combined score in a game | 100 / 370 |
| Who has the highest instagram followers in the world? | Are you asking about the individual with the most followers on Instagram, or the account with the most followers overall? | The individual with the most followers / The account with the most followers overall | Cristiano Ronaldo / Instagram |

Table 10: Randomly sampled Human-Ambiguity SFT dataset examples.

---

You will be given a question and several possible responses.
Possible responses may be correct due to an ambiguity in the question or when the question was asked.
Provide a clarifying question to determine which possible response is correct.
If one or zero of the possible responses are correct, respond with "None".
Provide your output in the following format:

Clarifying Question: [Clarifying Question]
1. Clarifying Answer: [First Clarifying Answer]
1. Response: [Copied Response for First Clarifying Answer]
2. Clarifying Answer: [Second Clarifying Answer]
2. Response: [Copied Response for Second Clarifying Answer]
...

Question: {question}
Possible Responses:
{answers}

---

Table 11: Prompt used for generating our Human and Model Ambiguity SFT datasets with GPT-4.

---

Pretend that you are a user asking an AI assistant a question.
In response to your question, the AI assistant has asked you a clarifying question to help it determine which answer you expect.
For each of the following expected answers, provide a clarifying answer to the AI assistant's clarifying question that indicates you are expecting that answer.
All clarifying answers should be a concise sentence or phrase and should not contain the expected answer.
If there is no possible clarifying answer, respond with "None." instead.
Provid your response in the following format:

Clarifying Answer 1: [Clarifying Answer for Expected Answer 1]
Clarifying Answer 2: [Clarifying Answer for Expected Answer 2]
...

Question: {question}
Clarifying Question: {clarify_q}
{answers}

---

Table 12: GPT-4 prompt used for simulating user clarifying answers during evaluation. All evaluations were carried out using the GPT-4 version gpt-4-0613.

Table 13: Example predictions where asking clarifying questions helped recover the correct answer to an unambiguous test query. Clarifying question predictions are taken from our LLama2-7B-based Clarify DPO (Ours) model trained with double-turn preferences. We also include the predicted LLama2-7B-based Direct-Ans SFT model predictions.

| Question ($x$) | Pred Clarifying Question ($q$) | GPT-4 Clarifying Answers ($a_i$) | Clarify DPO Pred Ans. ($\widehat{y_i}$) | Direct-Ans SFT Pred Ans. ($\widehat{y_i}$) | Gold Ans ($y_i$) |
|---|---|---|---|---|---|
| When does telltale's guardians of the galaxy episode 3 come out? | Which platform are you asking about for Telltale's Guardians of the Galaxy Episode 3? | I'm asking about the release date on all platforms. | August 22 , 2017 | October 17, 2017 | August 22 , 2017 |
| When did gods not dead 3 come out | Which "God's Not Dead 3" are you referring to? The first, second, or third installment? | I'm asking about the release date on all platforms. | March 30 , 2018 | April 2, 2018 | March 30 , 2018 |
| When did the sea of monsters movie come out? | Which "Percy Jackson and the Sea of Monsters" movie are you asking about? The one released in 2013 or the one released in 2022? | The one released in 2013. | August 7 , 2013 | March 26, 2010 | August 7 , 2013 |
| Who won the nobel prize for peace in 2014? | Which individual or organization won the Nobel Peace Prize in 2014? | The individuals who won | Kailash Satyarthi and Malala Yousafzai | Kailash Satyarthi | Kailash Satyarthi and Malala Yousafzai |
| How many islands are included in the federated states of micronesia? | Which specific islands are you referring to? | I'm referring to all the islands and islets that make up the country. | 607 islands and islet | 600 | 607 islands and islet |

Table 14: QA-with-Clarification predictions on unambiguous test queries from our LLama2-7B model trained with double-turn preferences. We include examples where clarifying questions helped the model recover multiple annotated answers.

| Question ($x$) | Pred Clarifying Question ($q$) | GPT-4 Clarifying Answers ($a_i$) | Pred & Gold Ans. ($\widehat{y_i}, \widehat{y_i}$) |
|---|---|---|---|
| How many starbucks ? are there around the world | As of which date are you asking about the number of Starbucks stores around the world? | As of the end of 2018. As of the end of 2004. As of the end of 2016. As of the end of 2017. | (28,218, 28,218) (6,000, 4,962) (23,768, 23,768) (25,000, 30,000) |
| Who sang the original do you love me? | Which version of "Do You Love Me" are you referring to? The original song by The Contours or the cover by The Partridge Family | The version that was a hit in the 1960s. The version from the musical Fiddler on the Roof. | (The Contours, The Contours) (Topol, Topol) |
| When did the queen became queen of england? | Which specific queen are you referring to? | Queen Victoria Queen Elizabeth II Queen Anne | (20 June 1837, 20 June 1837) (6 February 1952, 6 February 1952) (8 March 1702, 1 May 1707) |

