# OpenReview forum: "Modeling Future Conversation Turns to Teach LLMs to Ask Clarifying Questions"
_ICLR.cc/2025/Conference — ICLR 2025 Poster_

### Official Review · Reviewer_vGfa · 2024-10-25

**Soundness:** 2
**Presentation:** 1
**Contribution:** 3
**Rating:** 5
**Confidence:** 4

**Summary:**

This paper focuses on the problem of having LLMs ask clarification questions when the user request is ambiguous or uncertain. The authors propose a new approach to construct preference data by simulating future conversations. Experiments are conducted in two open-domain QA datasets. The results show that the proposed method outperforms other standard baselines.

**Strengths:**

1. This work studies an important question in the era of LLMs, i.e., teach LLMs to ask clarification questions
2. Propose a novel approach to construct preference data including clarifications by simulating future conversations
3. Experimental results show that the proposed method outperforms other standard baselines.

**Weaknesses:**

1. The presentation of the proposed method is a bit hard to follow. There is a lot of symbols and variables in the text, which is very messy and not well defined. More details in the questions that need to be clarified by the authors.
2. The information of annotators for preference data construction is also not clear.
3. The experimental settings are not extensive and reasonable enough, only comparing on self-designed baselines.
4. Several recent studies on LLM-based clarifications are not discussed or compared, such as [1-3].

Minors: The usage of \cite{} is incorrect.

[1] STaR-GATE: Teaching Language Models to Ask Clarifying Questions
[2] STYLE: Improving Domain Transferability of Asking Clarification Questions in Large Language Model Powered Conversational Agents
[3] Learning to Clarify: Multi-turn Conversations with Action-Based Contrastive Self-Training

**Questions:**

1. What is $a_i$ in line 120? It seems that it has a different meaning from ($a_i$) in line 129.
2. Are the input query $x$ and the user intent ($x_i$) supposed to be different forms?
3. Is the user-simulator model for preference data construction different from the the user simulation model in Section 2.1?
4. What not apply Clarify-or-Direct-Ans SFT as a baseline?
5. How is the original performance of these LLMs?
6. How about just using in-context learning for enabling the clarification capability?
7. Is there any existing method that can be applied as baselines, instead of all your self-designed baselines?

---

> ### Author Response · Authors · 2024-11-27
> **Author response**
>
> Thank you for your thoughtful response. We address each comment below, and provide additional details regarding added baselines in our general response above.
>
> **[W1] Writing clarity**
>
> Thank you for pointing out these points of confusion in your questions below. We address these points below and revise them in our updated manuscript.
>
> **[W2] Unclear Preference Data Construction**
>
> We do not collect new human annotations in this work. Instead, we use different annotator intents and their respective expected answers from NaturalQuestions dataset. We depict how these annotations are used for preference data construction in Figure 2. We use the annotations from NaturalQuestions to simulate the clarifying answers from each annotator. We then simulate each annotator’s preference by comparing the LLM’s predicted output against the respective annotator’s expected answer.
>
> **[W3-4] Additional Baselines and Related Works**
>
> In our related work, we discuss STaR-GATE and add it as one of our additional baselines in our general response above. There, we also add further discussion on these other cited works, and will include these in our revisions. Additionally, our baselines using the Starling Reward model to rank clarifying questions is used to demonstrate the efficacy applying standard RLHF methods in our clarifying questions setting.
>
> To compare against the original performance of these LLMs, in our General Response above, we add a comparison against using CoT prompting to evaluate the original performance of the base LLM. We find that CoT performance of the base LLM (post instruction-tuning) underperforms when compared to trained methods.
>
> **[Q1] Clarification on $a_i$ in lines 120 and 129**
>
> In both lines, $a_i$ is the $i$-th user’s clarifying question to the LLM’s proposed clarifying question. During evaluation, we simulate $a_i$ by conditioning on the input query $x_i$, clarifying question $r_{init}$, and the $i$-th user’s expected answer $y_i$.
>
> In line 120, there was a typo where we are $M(x)$ instead of $r_{init}$. The correct equation is represented in line 129 and in Figure 1. Thank you for pointing this out! We fix this in our revisions.
>
> **[Q2] Are the input query ($x$) and the user intent ($x_i$) supposed to be different forms?**
>
> The $x$ refers to the potentially ambiguous input query passed to the LLM. In Figure 1, we use $x_i$ illustrate $i$-th user’s disambiguated intent in the form of a re-written input query. $x_i$ is only used to help illustrate our example in Figure 1, and is not directly used in our methods or settings. Thanks for pointing out this point of confusion, we will remove the unnecessary notation to avoid confusion in our revisions.
>
>
> **[Q3-5] Additional Baselines.**
>
> Thank you for your suggestions. We address this in our response to [W3-4] above, where we discuss additional baselines provided in our general response.

---

### Official Review · Reviewer_kypX · 2024-10-31

**Soundness:** 3
**Presentation:** 2
**Contribution:** 3
**Rating:** 6
**Confidence:** 4

**Summary:**

This paper examines handling ambiguity, an often overlooked aspect of large language models (LLMs) in multi-round conversations with users. The authors first develop a pipeline to collect human preferences in a two-round setting and then apply Direct Preference Optimization (DPO) to several open-source LLMs to enhance their performance.

**Strengths:**

- The paper provides a thorough scientific investigation, building upon several interesting hypotheses about LLM responses by training (evaluating) on prior data (Line 17) and speculating about the differences between LLM ambiguity and human ambiguity (Line 305).
- The paper introduces a novel pipeline for eliciting human preferences in multiple rounds of conversation, even though I have some reservations regarding its operation.
-  The experiments are very thorough, considering different variations of LLMs. However, the study lacks comparisons with non-LLM methods.

**Weaknesses:**

## Weakness 1: Unclear Relationship Between Human Annotators and the User Simulator

The paper appears to ask humans to only grade the final prediction of the model (the final step as shown in Figure 2). It is unclear whether their grading aligns with their judgment of the clarifying questions. It would be more appropriate to set a predefined user goal at the beginning and have human annotators grade the final answer based on its consistency with the original goal and the arrangement of questions and answers in the intermediate steps.

I recommend that the authors refer to the conversational information-seeking literature in the IR community. For example, the paper [1] adopts a user simulator that maintains an initial item of interest (e.g., iPhone 16 128GB). During the question-answering stages, the user simulator responds to questions in line with the item's specifications (e.g., "How much storage do you want?" "128GB"). In this paper, it is unclear whether the human's final grading considers an initial goal and whether the grading assesses both the answer quality and adherence to that goal.

This is my major concern, and I hope the authors can provide further justification during the response period. Thank you!

## Weakness 2: Lack of Baselines Beyond LLM Fine-Tuning

While the paper includes numerous ablation studies and comparisons with other LLM tuning baselines, additional comparisons are needed to justify the method's effectiveness. I recommend the following:

- **Unsupervised Beam Search**: This approach allows LLMs to search the space of possible clarifying questions (implicitly) [2].
- **Chain-of-Thought**: Prompting LLMs to explicitly reason about possible ambiguities.
- **Non-LLM Baseline Methods**: Including methods for asking clarifying questions that do not rely on LLMs, as explored in various works [3, 4].

I hope the authors can try to add these results or explain why these methods are not suitable for comparison. Thanks!

## Weakness 3: Need for More Insightful Analysis

- **Knowledge Taxonomy**: A taxonomy for knowledge types is needed to make the dataset more insightful. What types of knowledge are more desirable for clarifying questions? If the original dataset does not provide this, consider latent word clustering or LDA?
- **Error Accumulation**: Correct me if I am wrong, but I do not see an analysis of how an incorrect clarifying question may lead to error accumulation in the final answer.

I hope the authors can discuss with me if these research questions are helpful and whether they plan to append them.

# References
1. [Conversational Recommender System](https://arxiv.org/abs/1806.03277) SIGIR 2018
2. [Self-Evaluation Guided Beam Search for Reasoning](https://arxiv.org/abs/2305.00633) NeurIPS 2023
3. [Asking Clarifying Questions in Open-Domain Information-Seeking Conversations](https://arxiv.org/abs/1907.06554) SIGIR 2019
4. [Asking Clarification Questions to Handle Ambiguity in Open-Domain QA](https://aclanthology.org/2023.findings-emnlp.772/) EMNLP 2023

**Questions:**

The presentation of the paper has significant room for improvement. I have identified several typos and areas for enhancement:

- **Line 16**: Can you specify whether it is *evaluated*? I think it should be *trained*, (or both)?
- **Line 212**: "using a we use a trained user-simulator model" – this sentence does not parse.
- **Lines 224–226**: The green box is confusing. Why are q1 and q2 being preferred simultaneously?
- **Line 186**: “by providing their answer the proposed clarifying question” – this sentence does not parse.
- **Line 206**: "in this work we use simulate user interactions for annotation." – this sentence does not parse.

**Generic Questions:**

- Is your human annotation IRB-approved?
- In the methods section, can you write the loss function for DPO? This would make your paper more self-contained and eliminate the need for readers to refer to the original DPO paper.
- Can you provide more details on the evidence supporting the hypothesis that model responses are evaluated only on their prior contexts (Line 17)? This is a very interesting hypothesis (perhaps the central hypothesis of the paper), so additional justification would be beneficial.
- Line 305 presents an interesting hypothesis: "We hypothesize that what is ambiguous for an LLM often deviates from what is ambiguous for humans." Has this hypothesis been proven?
- Can you specify the input format for the LLaMA model? If my understanding is correct, only the Chat version includes the system and user specifications.
- What is the computation time and the number of training samples used?


Building on all my comments, I'd vote for a 5/10 score now :)

---

> ### Author Response · Authors · 2024-11-27
> **Author Response P1**
>
> Thank you for your thoughtful review! Due to space constraints, we address comments here and will answer questions in the following response.
>
> **[W1] Unclear Relationship Between Human Annotators and the User Simulator**
>
> We do not collect human annotations ourselves for this work. Human annotations for different users' answers to each query  $(x, \{y_1, \dots, y_n \})$ are sourced from the AmbigQA and NaturalQuestions datasets. We use these annotations to simulate the clarifying answers $(a_1, \dots, a_n)$ from each human annotator to generated clarifying question $(q)$. This is done by training a user simulator model to that conditions on the input query, clarifying question, and human-annotated answer to simulate the annotator's clarifying answer $(x, y_i, q) \rightarrow a_i$.
>
>
> **[W2] Additional Baselines**
>
> Thank you for your suggestions. In our common response, we include additional baselines for points of comparison, and include additional discussion on methods from the other works you listed. While some methods are not immediately adaptable to our setting (e.g., they are designed to only work for retrieval-based QA systems), we include further discussion.
>
>
> **[W3] Need for More Insightful Analysis (Error Accumulation)**
>
> This is a fascinating point! While such an analysis is quite interesting, in practice we find that it is not always obvious when the system is asking an “incorrect clarifying question”. Consider the example from our LLama2 method below where the query is unambiguous and clarifying the question seems unnecessary.
> * Question: “When did gods not dead 3 come out?”
> * Gold Answer: “March 30, 2018”
> * Clarifying Question: “Which “God’s Not Dead 3” are you referring to? The first, second, or third installment?”
> * Clarifying Answer: “The third installment.”
> * Predicted Answer with Clarification: “March 30, 2018”
> * Direct Answer without Clarification: “April 2, 2018”
>
> Here, we find that seemingly extraneous clarifying questions can often improve model performance.
> Another way to determine whether a clarifying question is irrecoverably bad during evaluation is by using GPT-4. Recall that during evaluation, use GPT-4 as our clarifying-answer simulator and allow it to refrain from providing a clarifying answer if it judges that none exists. Using this as an indicator of an irrecoverably bad clarifying question, we can compute the percent of ambiguous questions where each system produces clearly erroneous clarifying questions. For example, we find that our 20% of our LLAMA2-based Clarify SFT system’s clarifying questions contain errors. We also find that with additional DPO training, this number is reduced to only 13%. In our revisions, we will include these additional figures and discussions for additional error analysis.

---

> > ### Author Response · Authors · 2024-11-27
> > **Author response (cont.)**
> >
> > Here, we address each of the remaining questions.
> >
> > **[Q1] Is your human annotation IRB-approved?**
> >
> > We do not collect human annotations ourselves for this work. Annotations for different user’s answers to each question are sources from the AmbigQA and NaturalQuestions datasets. We use these annotations to simulate the clarifying answers from each annotator. We then simulate each annotator’s preference by comparing the LLM’s predicted output against the respective annotator’s expected answer.
> >
> > **[Q2] In the methods section, can you write the loss function for DPO? This would make your paper more self-contained and eliminate the need for readers to refer to the original DPO paper.**
> > Thank you for your comment. We add this and further explanation of the DPO method in our revisions.
> >
> > **[Q3] Can you provide more details on the evidence supporting the hypothesis that model responses are evaluated only on their prior contexts (Line 17)?**
> >
> > We say that annotators must evaluate model responses based only on the prior context because of how preference annotation tasks are set up. It is standard practice in preference data collection to provide annotators with a conversation history and multiple sampled candidates for the next LLM response. Annotators are then tasked with ranking/scoring each candidate without ever interacting with the LLM system or any candidate responses. Annotators, therefore, only have access to the current candidate response and prior context when evaluating the response’s quality.
> >
> > In L40 we cite (Ouyang et al., 2022) the seminal work that started this preference annotation practice, and later in L42, we cite two widely used preference datasets that follow this practice. To the best of our knowledge, no publicly available preference dataset deviates from this standard annotation setup.
> >
> > **[Q4] Line 305 presents an interesting hypothesis: "We hypothesize that what is ambiguous for an LLM often deviates from what is ambiguous for humans." Has this hypothesis been proven?**
> >
> > We discuss this in our main results in the paragraph starting on Line 268. In our main results, we find that clarifying questions improves QA performance for both ambiguous and unambiguous questions, as determined by humans. These unambiguous questions are cases where humans determined that clarification is unnecessary, yet models still benefited from asking clarifying questions. This finding highlights that LMs may perceive these questions as ambiguous, while humans do not. We include examples of such questions and LM predictions in Table 12 in the Appendix.
> >
> > **[Q5] Can you specify the input format for the LLaMA model? If my understanding is correct, only the Chat version includes the system and user specifications.**
> >
> > As noted in L161, we simply prepend each line with the Turn / Chat Message. E.g., “Question: … \n Answer … \n” or “Question: … \n Clarifying Question: … \n Clarifying Answer: … Answer … \n”. We will further clarify this and include an example in the appendix in our revisions.
> >
> > **[Q6] What is the computation time and the number of training samples used?**
> >
> > Compute details are presented in Appendix B, where we note that all experiments were carried out on a single machine with 8 A40 (48GB) GPUs and all training runs completed within 24 hours. In systems description in Section 4.1, we state that we sample up to 6 unique clarifying questions per query, which resulted in ~95K preference pairs for training our Clarify DPO methods across all base models.

---

> > > ### Comment · Reviewer_kypX · 2024-12-01
> > > **Acknowledge your responses**
> > >
> > > Hi authors:
> > >
> > > Thanks for your clarifications on the user simulation. I increase my score by one point.
> > >
> > > Regards,

---

### Official Review · Reviewer_5Axj · 2024-11-02

**Soundness:** 3
**Presentation:** 2
**Contribution:** 4
**Rating:** 6
**Confidence:** 3

**Summary:**

This paper addresses the challenge of asking clarifying questions for ambiguous user queries. To determine the optimal timing and content of a clarifying question, one of the core contributions of this paper is a dataset of multi-turn QA dialogues that include such questions. The paper also presents a method for evaluating the quality of clarifying questions. In this method, multiple candidate clarifying questions are tested using a user simulator, with the quality of each question assessed based on the aggregate reachability to correct answers. Models trained on this dataset in various settings demonstrate improved QA accuracy on both ambiguous and unambiguous queries.

**Strengths:**

- Asking clarifying questions for ambiguous queries is a valuable trait for QA systems, and the paper’s proposed methods are well-motivated.
- The approach for assessing the quality of clarifying questions based on reachability to correct answers is novel, interesting, and well-reasoned.

**Weaknesses:**

- The primary weakness of this paper, in my opinion, is the lack of clarity in the method sections (2–4). One contributing factor is the use of terms without clear definitions. For example, in line 196, the term “annotators” is used, which seems to imply human annotators, while the preceding text suggests a “simulator” is involved. Additionally, the description of user simulation from line 208 onwards does not clearly explain the origins of *q* and *a_i*, making it challenging to follow. Although some of these ambiguities are clarified later in the paper, I would recommend reorganizing the structure to reduce the distance between concepts and their realizations. Further questions and suggestions are provided in the Questions section below.
- The paper contains numerous ungrammatical and broken sentences, which significantly impact readability. For instance, line 187 ("providing their answer the proposed"), line 191 ("catered a toward"), line 206 ("we use simulate user interactions"), line 212 ("using a we use"), line 300 ("examples SFT training"), line 307 ("to for queries"), and line 312 ("we a sample"), among others. Additionally, citations are incorrectly formatted on lines 087 and 097. I will raise my score if the authors thoroughly address the grammatical and formatting issues and their revisions in their rebuttal.
- Some experimental settings and design choices lack sufficient motivation. For example, why is asking clarifying questions preferable to simply providing multiple answers in one response (e.g., Q: "Where were the Olympic Games held in Greece?" → A: "The ancient Olympic Games were held in Olympia, and the modern Olympic Games were held in Athens.")? Moreover, there is no discussion on efficiency or how humans might perceive these clarifying questions, and a human evaluation might be valuable.
- There is no experiment involving training a model directly on multi-turn QA dialogues. Instead, the authors only experiment with combining the weights of two separate models. Since a multi-turn dataset is available, training and evaluating a model on this dataset should be feasible.
- In section 5.1, the comparison with the random baseline is questionable. A more effective comparison would evaluate whether the model asks clarifying questions when a query is ambiguous and refrains when unnecessary (e.g., based on the ambiguous and unambiguous splits of the dataset or cases where annotators provided consistent answers).

**Questions:**

- Line 164: Please clarify “We prompt GPT-4 to abstain from providing a clarifying answer a_i if it judges that none exists, in which case we count the resulting target answer prediction r_next^i as incorrect.”
- Line 245: What is “EM accuracy”?
- Lines 299: What does “if it determines one exists” mean and when can this happen?
- Line 312: Please clarify “generate its greedy answer with temperature T = 0.0 and sample an answer with T = 1.0”.
- Line 324: Which of the models below is used for the Turn 4 model?
- Lines 330—359: The purpose of each model and how they are used in the two-step pipeline is unclear.
- Line 342: What is the purpose of the Clarify DPO models?
- Line 347: More details about the reward model and the RL training are needed.
- Line 432: Why not training a model on the multi-turn dataset?
- Line 459: Why is the random baseline appropriate for comparison?
- Line 461: How did you set the value of DA?

---

> ### Author Response · Authors · 2024-11-27
> **Author Response**
>
> Thank you for your thoughtful comments, and thorough notes on how we can improve our writing. We will incorporate these suggested edits and upload them in our revisions. Below, we address each comment. Due to space constraints, we address questions in our following comment.
>
> **[W1, W2] Clarity of Methods Description in Sections 2-4**
>
> We provide these revisions in our updated draft. Below, we clarify a few of these points.
>
>  * Line 196: Our method uses expected answers from multiple human annotators to simulate each annotator’s interactions with the LLM and their preferences over different responses. Thus, we discuss other methods of using multiple human annotators in RLHF training here. These methods do not simulate interactions from these multiple human judgments.
> * Line 208:
> We clarify that these answer sets are sourced from multiple human annotators. We also provide a reference to our dataset descriptions (Section 4: Experiments) where the annotation setups are described in detail.
>
>
> **[W2] Grammatical issues**
>
> Thank you for identifying these errors! We have revised each in our updated draft.
>
> **[W3] Clarifying Questions versus Multi-Answer responses**
>
> We agree the task of determining when Clarifying Questions versus Multi-Answer responses are appropriate is an important line of inquiry. There have been some prior efforts [1] demonstrating that users prefer systems to ask Clarifying Questions responses over Multi-Answer responses about twice as frequently; however, there has been limited work analyzing when this is the case or building systems to determine which response is more appropriate. Ultimately, we agree that systems should be able to predict either response type based on what is most appropriate for the specific setting, user, and query. While this is an important line of inquiry for future work, here we focus solely on the task of generating clarifying questions.
>
> [1] Asking Clarification Questions to Handle Ambiguity in Open-Domain QA
> * Dongryeol Lee, Segwang Kim, Minwoo Lee, Hwanhee Lee, Joonsuk Park, Sang-Woo Lee, Kyomin Jung
> * EMNLP 2023 Findings
>
>
> **[W4] No experiment involving training a model directly on multi-turn QA dialogues. Instead, the authors only experiment with combining the weights of two separate models.**
>
> As we note in Section 5, model merging is an efficient alternative to training an additional model from scratch on the full multi-turn dialogues, and we expect only a minor gain in performance with training another model rather than merging (Wortsman et al., 2022). We also note recent works that propose methods that could yield even greater performance gains than training from scratch over the full dialogues. We chose to use parameter merging in our experiments for the following two reasons:
> We expect such improvements from training from scratch or using mixture-of-experts models to be orthogonal to the methods proposed in this work for assigning preferences over clarifying questions.
> The results from parameter merging are sufficient for demonstrating that performance gains over the Direct-Answer SFT baseline are not simply due to the increased parameter count from using separate models for each conversation turn, as merging to a single joint model still outperforms these methods.
>
> **[W5] Evaluation Metric in Section 5.1**
>
> In Section 5.1, our Direct-Answer Accuracy metric directly evaluates whether systems are able to ask clarifying questions when they are necessary and directly answer user queries when they are not. This metric differs slightly from simply evaluating binary classification accuracy of determining whether or not the query is ambiguous, as clarifying questions often helps systems recover correct answers to unambiguous questions as well (as demonstrated in the our main results in Table 1). To account for this, Direct-Answer Accuracy also rewards systems for asking clarifying questions for unambiguous queries if it would otherwise get the question wrong. We include examples of such instances Table 12.

---

> > ### Author Response · Authors · 2024-11-27
> > **Author Repsonse (Cont. 1)**
> >
> > We provide responses to questions here, and address the remaining questions in the following comment.
> >
> > **Line 164: Please clarify “We prompt GPT-4 to abstain from providing a clarifying answer a_i if it judges that none exists, in which case we count the resulting target answer prediction r_next^i as incorrect.”**
> >
> > Our prompt for simulating users’ clarifying answers (provided in Table 11) instructs GPT-4 to abstain from providing a Clarifying Answer for a given (Input Question, Clarifying Question, Expected Answer) pair if there is no Clarifying Answer that makes sense by generating “None” in lieu of of the Clarifying Answer. This happens when the clarifying question does not address the ambiguity in the query, hence the resulting expected answer predictions are considered incorrect during evaluation. Here is an example from our LLAMA3 Clarify DPO:
> > * Input Question: When is episode 113 of dragon ball super coming out for its original airdate?"
> > * Clarifying Question: Which episode of Dragon Ball Super are you referring to?
> > * Expected Answer: June 1, 2019
> > * GPT-4 Predicted Clarifying Answer: None
> > * Explanation: The question is ambiguous because the answer is different for the English and Japanese language releases, but the clarifying question does not address this. The expected answer in this case is the Japanese air date, but GPT-4 abstains from providing the clarifying answer since the question is not correctly addressing the ambiguity.
> >
> >
> > **Line 245: What is “EM accuracy”?**
> >
> > EM accuracy is the standard metric for evaluating Open-Domain established in [1]. This measures exact string match accuracy of the predicted answer against the gold after normalizing for casing, whitespace, etc. We will cite and clarify this in our revisions.
> >
> > [1] Reading Wikipedia to Answer Open-Domain Questions
> > * Danqi Chen, Adam Fisch, Jason Weston, Antoine Bordes
> > * ACL 2017
> >
> > **Lines 299: What does “if it determines one exists” mean and when can this happen?**
> >
> > See our response to Line 164 above, where we clarify this point.
> >
> >
> > **Line 312: Please clarify “generate its greedy answer with temperature T = 0.0 and sample an answer with T = 1.0”.**
> >
> > To generate a set of model-identified feasible answers, we select a 5-shot prompt and sample twice from the model to identify two possible answers: once with temperature T=0 (Greedy) and once with temperature T=1. We then repeat this process for multiple 5-shot prompt to generate the full model-identified feasible answers.
> >
> > **Line 324: Which of the models below is used for the Turn 4 model?**
> >
> > For our experiments with each base model (Llama-2, llama-3, Gemma), we finetune the same base model to perform Turn 4. This model is finetuned on our SFT dataset, which we depict in Figure 3.
> >
> > **Lines 330—359: The purpose of each model and how they are used in the two-step pipeline is unclear.**
> >
> > We depict how all systems are trained and used in Figure 3. Our two-step training pipeline follows the standard (1) SFT training and (2) learning-from-preferences steps. Our Clarify SFT model is the system after this first step in this training pipeline. Our Clarify DPO models then complete the second step of training, but are trained with different methods for generating preferences over clarifying questions. For our Clarify-or-Direct Answer DPO model, we again perform this second training step, but include preferences over both clarifying questions and direct-answer responses.
> >
> > **Line 342: What is the purpose of the Clarify DPO models?**
> >
> > The Clarify DPO models are systems that always ask clarifying questions for each input question. We use these systems to directly compare the clarifying question generation ability of different systems trained with [1] standard single-turn preferences via Starling, [2] our proposed double-turn preferences, and [3] STaRGATE preferences (introduced in our general response above). For fair comparison, we use the same DPO training algorithm and clarify all input questions in these experiments.
> > Line 347: More details about the reward model and the RL training are needed.
> > In lieu of training a reward model and training via PPO, we train using DPO. This popular alternative optimizes the same objective as PPO, but does not require training a separate reward model. We will add this and further explanation of the DPO method in our revisions for clarity.

---

> > > ### Author Response · Authors · 2024-11-27
> > > **Author Response (Cont. 2)**
> > >
> > > We address the remaining questions below:
> > >
> > > **Line 432: Why not training a model on the multi-turn dataset?**
> > >
> > > In our main results (Table 1), we compare different methods for training models to ask clarifying questions. To ensure that differences in performance are only a reflection of the quality of the generated clarifying questions from each system, we use separate models for each conversation turn, keeping the answer prediction (Turn 4) model the same across all compared clarifying-question generation (Turn 2) methods. We later explore developing a joint, multi-turn model in Lines 430-450 using parameter merging. We discuss this further in our response to [W4] above.
> > >
> > > **Line 459: Why is the random baseline appropriate for comparison?**
> > >
> > > We use the random baseline to demonstrate that our Clarify-or-Answer DPO is able to learn to determine when to ask a clarifying question, but also that this task remains very difficult. We include additional baselines in our General Response above.
> > >
> > > **Line 461: How did you set the value of DA?**
> > >
> > > The value of DA is set by the trained, clarify-or-answer systems' predictions. After evaluating the performance of our clarify-or-answer DPO system, we compare against the random baseline of randomly selecting the same percent of questions (DA) to directly-answer versus clarify, thus matching the overall average conversation turn length.

---

> > > > ### Comment · Reviewer_5Axj · 2024-12-03
> > > >
> > > > Thank you for your responses and clarifications.
> > > > Most typos and broken sentences seem to have been fixed, although there are still some remaining issues (as in line 344). I have raised my scores.

---

### Official Review · Reviewer_UjMF · 2024-11-04

**Soundness:** 3
**Presentation:** 2
**Contribution:** 3
**Rating:** 5
**Confidence:** 4

**Summary:**

The paper proposes an approach which constructs preference data which are determined by examining (up to 2) simulated future dialogue turns. The proposed approach is evaluated in an open-domain QA dataset, NaturalQuestions, and a related derivative dataset, AmbigQA. The authors compare their approach against a standard single-turn reward model and present their results in terms of Token-based F1 for the QA task. The authors also present a set of experiments on ambiguity recognition.

**Strengths:**

***Significance of setup***

The proposed setting is important to investigate. As LLMs are increasingly tuned for use in dialogue applications it is important to consider how to better address the needs of users, and disambiguation is certainly a reasonable idea.

***Ablations and analysis***

The paper presents experiments comparing the proposed approach on multiple base LLMs, and performs multiple reasonable ablations. The paper also considers and interesting examination on the effect of obtaining human annotations versus model predictions.

***Clarity***

Overall the paper is clear and the experimental details are helpful.

**Weaknesses:**

***Concerns regarding generalizability***

The focus and main contribution of the paper is that it considers a lookahead approach for future conversation turns in order to optimizing preferences in question-answering settings. However, the approach is only defined for at most two future conversation turns. It is not clear how 1. this may be generalized to longer rollouts and 2. whether this approach would remain effective in such a setting.

***Scope of experiments***

Relatedly, the domains examined are limited. The approach is validated on NaturalQA and AmbigQA. AmbigQA is an extension of NaturalQA, and as a result it is not clear nor trivial how the proposed approach would generalize to any other more complex and realistic domains (e.g., those requiring complex rollouts, or task-oriented settings).

***Limited baselines***

The baselines in the main experiments are somewhat limited. If I understand correctly, Direct Answer SFT and Clarify SFT both do not seem like fair baselines in a task where there is mixed ambiguity. How does the proposed approach compare against SFT on both types of responses? Moreover, the proposed approach focuses on using RL tuning combined with future turns. How does it compare against directly including those future turn trajectories? It would also be helpful to compare against other existing RL-based approaches for ambiguity detection/action planning (e.g. [1]).

There are also no baselines for the ambiguity recognition experiments. Does the proposed approach outperform prompting? The objective performance of the proposed model here seems rather low. If the model asks questions unnecessarily, it may lead to degraded user experience even if it successfully elicits additional information to solve the downstream task / provides additional context to condition on for answer generation.

*Minor Points*

typos:
L187 "their answer the proposed"
L300 "construct all examples SFT training"
L306 "to for queries"

References

[1] Plug-and-Play Policy Planner for Large Language Model Powered Dialogue Agents, ICLR 2024

**Questions:**

1. How do you consider scenarios in which clarifying questions are undesirable behavior? Relatedly, how do you propose to balance clarifying questions with behaviors such as hedging, or presenting information for all possible intents?

2. Does the motivating problem of LLMs not asking clarifying questions get solved by prefixing with a simple instruction such as "ask a clarifying question if ambiguous"? For instance, I tried this with the example from Figure 1 and Gemini asks a clarifying question. To that end, such systems may be tailored to one type of behavior over another by design?

3. Have you considered multiple types of ambiguity (e.g. contextual versus lexical)?

4. Is the proposed approach intended to improve general LLM usage, or for this one particular task? if the latter, is this task with basic response outputs (e.g. single token) representative of real-world usage?

5. L161 is the model specifically fine-tuned to output "Clarifying Question:" (i.e. all generated responses should be accompanied by an intermediate dialogue act token)?

---

> ### Author Response · Authors · 2024-11-27
> **Author Response**
>
> Thank you for your thoughtful comments. We address comments below and questions in the following commment.
>
> **[W1&2] Generalizability to multi-turn setting + Scope of experiments**
>
> We agree that extending our evaluation and methods to arbitrary conversation lengths is an exciting avenue for future work. We select this QA setting for two reasons: (1) QA is one of the most prevalent request types in real user-LLM interactions [1] and (Zhao et al. 2024) and (2) the scope of possible clarifying questions is unbounded (prior works, as we note in L107-110 in Section 2 and L498-510 have focused on grounded settings where possible ambiguities are identified within the provided context or fixed by the task.). This setting, however, often does not necessitate more than two conversation turns to resolve the ambiguity in the request  (see response to Q3 below for an analysis of different ambiguity types). To the best of our knowledge, we are the first Future work may explore extending our settings requiring more conversation turns.
>
>
> [1] LMSYS-Chat-1M: A Large-Scale Real-World LLM Conversation Dataset
> * Lianmin Zheng, Wei-Lin Chiang, Ying Sheng, Tianle Li, Siyuan Zhuang, Zhanghao Wu, Yonghao Zhuang, Zhuohan Li, Zi Lin, Eric P. Xing, Joseph E. Gonzalez, Ion Stoica, Hao Zhang
> * Arxiv 2023
>
> **[W3] Limited Baselines**
>
> Thank you for your suggestions! In our general response, we add such baselines including the PPDPP method from “Plug-and-Play Policy Planner for Large Language Model Powered Dialogue Agents” you suggested and include it as an additional baseline. Overall, the new baseline achieves slightly lower performance than our proposed method. See the general response above for further discussion.
>
> Regarding your comments on the low Direct-Answer Accuracy performance of clarify-or-answer systems, we discuss its weakness in Section 5.1. In our General Response above, we additionally include the binary classification accuracy comparing each system’s clarify vs. direct-answer prediction against the human-annotated ambiguous vs. unambiguous labels from AmbigQA. Note that PPDPP is the only method that directly utilizes human-annotated labels during training, and archives the best accuracy using this metric. Furthermore, note that this performance does not directly correlate to performance on our end-task F1 metric, while our original Direct-Answer Accuracy metric does. Our Direct-Answer Accuracy metric is slightly different than ambiguity detection, as it directly measures whether systems are predicting to ask clarifying questions where they will improve the model’s response, also considering the cases where (1) asking a clarifying question for an unambiguous question improves the model’s response and (2) asking a clarifying question for an ambiguous question does not improve the model’s response. As we note in our introduction and results sections, our main results in Table 1 demonstrate that clarifying questions frequently improve model performance on ambiguous questions, which points toward discrepancy in what humans and models often identify as ambiguous.

---

> > ### Author Response · Authors · 2024-11-27
> > **Author Response (cont.)**
> >
> > We address remaining question below.
> >
> > **[Q1] How do you consider scenarios in which clarifying questions are undesirable behavior? Relatedly, how do you propose to balance clarifying questions with behaviors such as hedging, or presenting information for all possible intents?**
> >
> > We do not consider such scenarios, but discuss them as Future Work in Section 7. Hedging, multi-answer responses, and clarifying questions each represent reasonable dialogue acts for responding to an ambiguous request. Future work might explore methods for predicting when users may see one option as more appropriate than others.
> >
> > **[Q2] Does the motivating problem of LLMs not asking clarifying questions get solved by prefixing with a simple instruction such as "ask a clarifying question if ambiguous"? For instance, I tried this with the example from Figure 1 and Gemini asks a clarifying question. To that end, such systems may be tailored to one type of behavior over another by design?**
> >
> > We experiment with such promoting baselines in our General response above. Overall, we find that chain-of-thought prompting approaches perform worse than our proposed system. We agree though that developing and evaluating adaptable systems that can change their clarifying vs direct answering behaviors at test time is an interesting direction for future work.
> >
> >
> > **[Q3] Have you considered multiple types of ambiguity (e.g. contextual versus lexical)?**
> >
> > The datasets explored in this work cover a range of ambiguities. The authors of the original dataset perform the following analysis to identify the range of different ambiguities in the AmbigQA and NaturalQuestions datasets and their frequencies.
> > Event references (39%): The question contains an ambiguous even reference that could refer to multiple referents.
> > Entity references (23%): The question contains an ambiguous named entity reference that could refer to multiple referents.
> > Properties (27%): Answering the question requires some additional details.
> > Answer Types (16%): The query does not specify what answer type the user is looking for (e.g., what units of measurement).
> > Time Dependency (13%) The answer to the question changes over time.
> > Multiple Sub-Questions (3%): The question contains numerous sub-questions.
> >
> > **[Q4] Is the proposed approach intended to improve general LLM usage, or for this one particular task? if the latter, is this task with basic response outputs (e.g. single token) representative of real-world usage?**
> >
> > Great question! Our approach and evaluation metric is intended to be able to be extended to general LLM usage. To reflect this, our description of our LM evaluation framework in Section 2.1 is task-agnostic, and we describe the implementation + dataset details for our QA setting in Section 2.2.
> >
> > Having said that, there will be challenges for adapting it to wider domain: (1) collecting a dataset for the new setting with multiple annotator judgments and (2) changing evaluation metrics answer-match metrics to another task specific metric (e.g., BLEU) or to a generic task-agnostic evaluation metrics (e.g., LLM-as-Judge scores). We clarify this point and add this to our discussion of future work (section 7).
> >
> > **[Q5] L161 is the model specifically fine-tuned to output "Clarifying Question:" (i.e. all generated responses should be accompanied by an intermediate dialogue act token)?**
> > Yes, we include prepend “Clarifying Question:” to all clarifying question outputs during both SFT and DPO training.

---

> > > ### Comment · Reviewer_UjMF · 2024-12-02
> > >
> > > Thank you for your detailed response, and for your additional experiments. The additional comparison against PPDPP in the revised paper is helpful and I have increased my soundness score. However, my more pressing issue regarding the generalizability of the method and the scope of the experiments has not been addressed. In particular:
> > >
> > > > "We agree that extending our evaluation and methods to arbitrary conversation lengths is an exciting avenue for future work. We select this QA setting for two reasons: (1) QA is one of the most prevalent request types in real user-LLM interactions [1] and (Zhao et al. 2024) and (2) the scope of possible clarifying questions is unbounded (prior works, as we note in L107-110 in Section 2 and L498-510 have focused on grounded settings where possible ambiguities are identified within the provided context or fixed by the task.)" ... "Our approach and evaluation metric is intended to be able to be extended to general LLM usage. To reflect this, our description of our LM evaluation framework in Section 2.1 is task-agnostic, and we describe the implementation + dataset details for our QA setting in Section 2.2."
> > >
> > > I fully agree that QA is an important task and that many user requests can be considered questions. I do not have doubts about the significance of this overall topic. My point I made in my initial review was that AmbigQA is derived from NaturalQA, thus leading to similar user request distributions. Browsing through the AmbigQA questions, many of them are stylistically similar. I am concerned whether success on this dataset will translate to other QA datasets, or requests beyond toy datasets. I realize that it is quite late in the discussion period at this point, but given that the goal is to propose a "task-agnostic" approach, I think future versions of the paper would be strengthened greatly by: a) demonstrating Clarify DPO's efficacy on other datasets, b) directly demonstrating performance improvements in a representative sample of user requests (e.g., with human evaluation), or some combination of the two.
> > >
> > > >[Q1] How do you consider scenarios in which clarifying questions are undesirable behavior? ... Response: "Future work might explore methods for predicting when users may see one option as more appropriate than others."
> > >
> > > In Table 1, when Clarify-or-Direct-Ans DPO performance improves on Answer F1 while Average Turns also increases (perhaps undesirably?) Intuitively, additional clarification turns should strictly improve performance as there is additional context to form an answer, but additional clarification turns may not be desirable. It would be important to further investigate this point and discuss this in the paper.
> > >
> > > ---
> > >
> > > Side points:
> > >
> > > > "In our general response, we add such baselines including the PPDPP method from “Plug-and-Play Policy Planner for Large Language Model Powered Dialogue Agents” you suggested and include it as an additional baseline. Overall, the new baseline achieves slightly lower performance than our proposed method. See the general response above for further discussion."
> > >
> > > I'm unable to see a general response; did you set the Readers for it correctly?
> > >
> > > > To the best of our knowledge, we are the first Future work may explore extending our settings requiring more conversation turns.
> > >
> > > Sorry could you expand on what you are trying to say here?
> > >
> > > ---
> > >
> > > By the way, although this does not affect my score, I think Reviewer 5Axj raises an important point: "There is no experiment involving training a model directly on multi-turn QA dialogues. Instead, the authors only experiment with combining the weights of two separate models. Since a multi-turn dataset is available, training and evaluating a model on this dataset should be feasible."
> > >
> > > It seems like a reasonable baseline to directly tune a model on multi-turn dialogue. The conditions that souping has previously been used for are pretty different from multi-turn QA, to my knowledge. Overall, it somewhat feels like the MoE line in this paper distracts from the main point of the work.

---

> > > > ### Author Response · Authors · 2024-12-03
> > > > **Replying to Reviewer Response**
> > > >
> > > > Thank you for your prompt response. We address the remaining concerns below.
> > > >
> > > > **[W1] Generalizing to other QA datasets.**
> > > >
> > > > We agree that experiments on additional QA datasets would strengthen this work. However, we could not find other suitable dataset as they should come with multiple annotations to support our study. Most QA datasets (e.g., TriviaQA, HotpotQA, etc….) do not provide that.
> > > >
> > > > **[Q1] How do you consider scenarios in which clarifying questions are undesirable behavior? Intuitively, additional clarification turns should strictly improve performance as there is additional context to form an answer, but additional clarification turns may not be desirable.**
> > > >
> > > > We consider and evaluate this aspect of unnecessary clarifying questions as undesirable behavior in our experiments in Section 5.1 (Table 4). We capture this with the “Direct-Answer Accuracy” metric, though we have realized that the term is confusing (and will change in our revised version).
> > > > More intuitive name for this metric would be “Accuracy on deciding when to ask clarifying questions”, evaluating each system binary decision on whether to “Directly-Answer” vs. “Ask a Clarifying Question”. To compute this metric, for each example, we simulate the model's final answer set with the “Directly-Answer” option and “Ask a Clarifying Questions”. If the clarifying question improves the Answer F1 of the system’s prediction, then the gold label is to “Ask a Clarifying Question”. If the clarifying question does not improve the Answer F1, then the clarifying question is extraneous and the gold label is to “Directly-Answer”. Intuitively, method will be rewarded if (1) they asked clarifying questions and scored better than directly answering, or (2) they did not ask clarifying questions and scored as well as asking clarifying questions. This metric directly punishes systems that ask extraneous additional clarification turns.
> > > > We can see our method outperform other methods that make such Directly-Answer vs. Ask a Clarifying Question decision, such as PPDPP and Chain-of-Thought Prompting. We will make this clearer in the revised version.
> > > >
> > > > **Side Point 1: Missing General Response**
> > > >
> > > > The general response should be visible now, we apologize for the confusion.
> > > >
> > > > **Side Point 2: Explain “To the best of our knowledge, we are the first Future work may explore extending our settings requiring more conversation turns.”**
> > > >
> > > > Apologies for the confusion, that sentence in our prior comment intended to read that “Future work may explore extending our settings requiring more conversation turns.”

---

### Official Review · Reviewer_jxRi · 2024-11-11

**Soundness:** 3
**Presentation:** 3
**Contribution:** 3
**Rating:** 8
**Confidence:** 2

**Summary:**

The paper proposes a method for training LLMs in question-answering tasks to learn to ask clarification questions where appropriate given query ambiguity.

**Strengths:**

The topic addressed -- clarification behaviour by LLMs, and how to make their behaviour more accurate & reliable by clarifying where required -- is very topical and of great interest to the NLP community.

The paper proposes a fairly simple and intuitive method, describes it clearly, and evaluates it sensibly (mostly -- see below), and results seem good in that they show reliably increased accuracy.

**Weaknesses:**

The evaluations and baselines compared to could be stronger or more meaningful in some cases. In the main experiment, one of the main takeaways from the results seems to be that answer accuracy is increased if a CQ is always asked, although this comes at the cost of more turns. This raises the question of how well a system would do if given a very simple fixed prompt which forced a CQ as the first response, and then an answer based on the resulting three-turn context - but without going through the extra training steps given here. I expect it wouldn't be quite as good as the best system here, but it would be very instructive to know how much difference there was.

In the experiment of sec 5.1 (determining whether CQs are necessary), performance is compared to a random baseline (sampling a random fixed percentage of direct answers vs CQs), but there's no direct evaluation of the main question here: how often a CQ was asked after an ambiguous question (i.e. accuracy of identifying when a CQ is required). In fact, the question is perhaps better: how often was a CQ asked after a question which was ambiguous, and where the possible answers differ (if the answers are the same, a CQ provides no benefit, even if the question is nominally ambiguous).

The data on possible CQs and answers, including answer correctness/preference, is mostly drawn from LLM-based simulations, but there doesn't seem to be any validation of how accurate or realistic this is.

There's no discussion of how ambiguity in a question actually relates to differences in answers. Ambiguity sometimes doesn't matter in practice: e.g. a variant of the first question in Table 8 (appendix) could be "what country were the first olympic games held in"; then, although there is ambiguity about whether the query concerns the modern or ancient games, that doesn't affect the correct answer which would be Greece in both cases. It would be interesting to know how common this is, and whether the behaviour of the models reflects (or should reflect) this difference.

The biggest weakness I perceived is the lack of relation to other relevant work. There's discussion of related work with LLMs and ambiguity, and LLMs being trained to ask CQs in different contexts; but there's no discussion of pre-LLM work that seems relevant. For instance, there is a large body of work in training dialogue systems to respond suitably to ambiguous input (including asking clarification questions), much via reinforcement learning. Closer to the domain here, if usually less close in terms of method, is the body of work in interactive information retrieval (including using clarifying questions to refine or further specify search queries or QA questions). Some discussion of how the method here resembles (or doesn't resemble) those would be very helpful.

**Questions:**

Would it be helpful & possible to include some more informative comparisons and evaluation metrics (see "Weaknesses" above)?

On annotation reliability: I wasn't clear whether the additional annotation described for AmbigQA was done as part of this work, or previously by other authors and described elsewhere ("his process identifies about 56% of all queries in NQ-Open as ambiguous" etc.).

Table 2 doesn't say what metric it uses - presumably F1?

Some minor typos:
083 two axis->axes
097 citation should be citep not citet
110 of [the] task scenario
187 an additional turn[s]
406 Cla[i]rifying
188 observ[ing] the LLM's
206 simulate[d] user

---

> ### Author Response · Authors · 2024-11-27
> **Author Response**
>
> Thank you for your thoughtful remarks! We address your comments individually below.
>
> **[W1] Additional Baselines:**
> Thank you for your suggestions! Please see the general response where we add and discuss several additional baselines.
>
> **[W2] Section 5.1 (When to Clarify) Accuracy Metric:**
> In Section 5.1, our metric for Direct Answer accuracy (DA Acc) is intended to measure exactly this: how often are clarifying questions asked when they help LLMs recover more correct answers to the question. We chose to evaluate this accuracy metric rather than accuracy on identifying ambiguous/unambiguous questions because, as our main results in Table 1 demonstrate, clarifying questions not only help LLMs recover correct answers to ambiguous questions, but also for unambiguous questions as well. As such, this accuracy metric has the additional benefit of also rewarding models for asking questions in response to unambiguous questions if it helps the model recover the correct answer. We include examples of such questions in Table 12.
>
>
> **[Q1] Would it be helpful & possible to include some more informative comparisons and evaluation metrics (see "Weaknesses" above)?**
>
> See our discussions of weakness above where we address adding such additional comparisons.
>
> **[Q2] Human Annotation Details and Reliability**
> All annotations (both identifying ambiguous questions, and identifying the set of answers from different interpretations) were collected by the authors of the original AmbigQA work. We state that the annotations are sourced from the original datasets in Section 4, and clarify this in earlier mentions.
>
> **[Q3] Table 2 doesn't say what metric it uses - presumably F1?**
> Yes, we use Answer F1. We added this to the caption of Table 2.
>
> **[Q4] Typos**
> We have fixed the typos in the updated manuscript. Thank you for catching them.

---

### Comment · Reviewer_kypX · 2024-11-22
**Shall we engage in author response?**

Dear AC, my colleague reviewers,

I got an auto email reminder that we should engage in author response. When I checked in OpenReview, I didn't see anything.

Have you guys seen the author responses?

Regards,

---

### Author Response · Authors · 2024-12-02
**General Response**

We thank reviewers for their feedback and suggestions. We address individual reviewer’s comments separately, and report additional baselines which were suggested by multiple reviewers. In the following comment, we discuss and compare other related works noted by reviewers.

**Discussion of Baselines:** Thank you for helpful suggestions! We added results from several other relevant baseline methods, suggested by reviewers (Table 1 in the updated pdf), and discuss them below.

1. *STaR-GATE: Teaching Language Models to Ask Clarifying Questions Arxiv 2024:* Similarly to our method, they evaluate clarifying questions based on their expected reward. Here, clarifying questions are ranked by the likelihood of generating the gold answer after observing each simulated user’s clarifying answer, averaged over all simulated users. In contrast, we rank clarifying questions based on the accuracy of the predicted answer after observing each simulated user’s clarifying answer, averaged over all simulated users. However, their model is capable of only comparing different clarifying questions, and does not allow models to determine whether or not clarification is necessary.\
\
However, their model is capable of only comparing different clarifying questions, and does not allow models to determine whether or not clarification is necessary.\
\
To directly compare their method, we follow the same training procedure (same training response pairs, base model, etc) as our other models and only differ in the ranking of clarifying questions at preference learning. We find our method significantly outperforms STaR-GATE baseline. We hypothesize that may be due to the STaR-GATE methods’ sensitivity to generation likelihoods. When presented with two possible clarifying questions where neither are helpful, the minor variation in the gold-answer’s likelihood will result in STaR-GATE assigning preferences over the pair of clarifying questions. In contrast, our method can successfully identify in these cases that both clarifying questions are unhelpful, and will rank both questions as tied.

2. *Plug-and-Play Policy Planner for Large Language Model Powered Dialogue Agents ICLR 2024:* Similar to our work, this model (PPDPP) decides whether to ask clarifying questions or directly answer. However, they achieve this by training a separate LLM to predict whether or not to ask a clarifying question, while fixing the LLM which generates clarifying questions or answers. Our method has a single model for both clarifying question vs. answer decisions and generating the actual answers. We find their method performs slightly worse than us in terms of classifying when clarifying questions is necessary, but performs competitively overall at the significant cost of requiring a finetuning a separate LLM to decide when to clarify vs. directly respond to each test input.

3. *Prompting and Evaluating Large Language Models for Proactive Dialogues: Clarification, Target-guided, and Non-collaboration EMNLP 2023 Findings:* Here, we experiment with using a proposed Chain-of-Thought prompting method (ProCoT) to task models with (1) identifying whether or not the input question is ambiguous and (2) Generating a clarifying question or a direct-answer response if they if the question is ambiguous. We find that for these experiments, using an instruction-tuned base model is necessary for generating non-degenerate outputs. These results may, therefore, be slightly inflated due to test-train overlap. Despite this, we see that CoT prompting does worse than all other baselines.

4. *Clarify-or-Answer SFT:* Reviewers vGfa asked for Clarify-or-Answer SFT model, prior to DPO training, we include the results here. We find that, prior to DPO training, the Clarify-or-Answer SFT model learns to simply directly answer all input queries. This is most likely due to the data imbalance, which contains over 2 times more direct answers than clarifying questions. During DPO training, however, systems are able to learn to judiciously determine when to ask clarifying questions.


We report these results using LLaMa2 as a base model for computational resources limitation during the rebuttal period. We will include full results on different base models in our further revisions. We do not anticipate the trends to change.

---

> ### Author Response · Authors · 2024-12-02
> **General Response (cont.)**
>
> **Additional Related Works:** Reviewers suggested several other methods. Below, we include discussions for several of these methods, and discuss their similarities.
>
> * *Learning to Clarify: Multi-turn Conversations with Action-Based Contrastive Self-Training, ArXiv 2024*
> Suggested by vGfa. While this method’s training approach similarly simulates conversation and to rank responses for DPO training, their proposed method relies on access to gold conversation trajectories during RLHF training and evaluation. Such trajectories are not available in the NaturalQuestions dataset.
>
> * *STYLE: Improving Domain Transferability of Asking Clarification Questions in Large Language Model Powered Conversational Agents ACL Findings 2024, Asking Clarification Questions to Handle Ambiguity in Open-Domain QA EMNLP Findings 2023*
> Suggested by vGfa and kypX. These two methods are designed for the RAG setting where each question is also supplemented with a set of retrieved documents paired with their retrieval/relevance scores. In this work, we focus on the closed-book setting.
>
> * *Self-Evaluation Guided Beam Search for Reasoning NeurIPS 2023.*
> Suggested by kypX. This work presents a decoding method to achieve better accuracy and calibration with Chain-of-Thought prompting. While it can allow for models to consider different intermediate / chain-of-thought-style questions, it doesn’t enable interaction.
>
> * *Asking Clarifying Questions in Open-Domain Information-Seeking Conversations SIGIR 2019*
> Suggested by kypX. This early work explores the task of asking clarifying questions in similar open-domain clarifying question settings. The methods in this work, however, rely on selecting from a fixed pool of existing clarifying questions, severely limiting their applicability.

---

### Meta-Review · Area_Chair_MNa8 · 2024-12-22

**Metareview:**

This paper proposes to ask clarifying questions to the users to elicit more information from them in the case of ambiguous questions for LLM-based question answering. Experimental results show 5% improvement in F-measure with an LLM fine-tuned to ask clarifying questions. Reviewers highlight the simplicity and meaningfulness of the idea as the strength of the work. The weaknesses listed include many questions that aim to clarify experiments and decisions made in the work, and its positioning with respect to earlier work.

**Additional Comments On Reviewer Discussion:**

Rebuttal period included several discussions between the authors and reviewers, and resulted in some of them increasing their scores based on the additional experimentation and clarifications provided by the authors.

---

### Decision · Program_Chairs · 2025-01-22

Accept (Poster)